# One Model for All: Multi-Objective Controllable Language Models

**Qiang He**                                              *Qiang.He@ruhr-uni-bochum.de*
*Ruhr University Bochum*

**Yucheng Yang**                                               *y.yang@tue.nl*
*Eindhoven University of Technology*

**Tianyi Zhou**                                        *tianyi.zhou@mbzuai.ac.ae*
*Mohamed bin Zayed University of Artificial Intelligence*

**Meng Fang**                                          *Meng.Fang@liverpool.ac.uk*
*University of Liverpool*

**Mykola Pechenizkiy**                                    *m.pechenizkiy@tue.nl*
*Eindhoven University of Technology*

**Setareh Maghsudi**                               *Setareh.Maghsudi@ruhr-uni-bochum.de*
*Ruhr University Bochum*

**Reviewed on OpenReview:** *https://openreview.net/forum?id=qAM5PmvFYY*

## Abstract

Aligning large language models (LLMs) with human preferences is critical for enhancing LLMs' safety, helpfulness, humor, faithfulness, etc. Current reinforcement learning from human feedback (RLHF) mainly focuses on a fixed reward learned from average human ratings, which may weaken the adaptability and controllability of varying preferences. However, creating personalized LLMs requires aligning LLMs with individual human preferences, which is non-trivial due to the scarce data per user and the diversity of user preferences in multi-objective trade-offs, varying from emphasizing empathy in certain contexts to demanding efficiency and precision in others. *Can we train one LLM to produce personalized outputs across different user preferences on the Pareto front?* In this paper, we introduce M̲ulti-O̲bjective C̲ontrol (MOC), which trains a single LLM to directly generate responses in the preference-defined regions of the Pareto front. Our approach introduces multi-objective optimization (MOO) principles into RLHF to train an LLM as a preference-conditioned policy network. We improve the computational efficiency of MOC by applying MOO at the policy level, enabling us to fine-tune a 7B-parameter model on a single A6000 GPU. Extensive experiments demonstrate the advantages of MOC over baselines in three aspects: (i) controllability of LLM outputs w.r.t. user preferences on the trade-off among multiple rewards; (ii) quality and diversity of LLM outputs, measured by the hyper-volume of multiple solutions achieved; and (iii) generalization to unseen preferences. These results highlight MOC's potential for real-world applications requiring scalable and customizable LLMs.

## 1 Introduction

Large language models (LLMs) have gained significant attention for their impressive performance across a wide range of tasks, including machine translation (Vaswani et al., 2017; Radford & Narasimhan, 2018; Devlin et al., 2019), text generation (Touvron et al., 2023; OpenAI, 2023), and conversational agents (Ouyang et al.,

2022; Bai et al., 2022). However, these models are generally aligned with fixed preferences predetermined by developers (Ouyang et al., 2022; Touvron et al., 2023; Bai et al., 2023; Dubey et al., 2024), limiting the degree of personalization available to users. In real-world scenarios, users often have diverse preferences for LLMs behaviors. For example, individuals often seek varying trade-offs between traits like empathy and task-oriented efficiency depending on the immediate context. Despite this variability, the inherent flexibility of current LLMs (Dubey et al., 2024; OpenAI, 2023) is limited in adjusting to fine-grained, personalized interaction preferences (Guan et al., 2025; Li et al., 2025).

The ability of LLMs to dynamically balance the trade-offs between different objectives according to diverse user-specified preferences is called *multi-objective controllability* (Guan et al., 2025; Li et al., 2025), a crucial feature for enhancing user satisfaction. Training separate models for each preference order, however, is neither practical nor scalable due to the high computational costs associated with LLM training. This highlights the need to enable one-time LLMs training while accommodating a broad range of preferences.

Table 1: Comparison with the state-of-the-art MOO methods. MOC addresses MOO in a principled and efficient manner. $M-$ the number of preferences, $N-$ the number of reward models (objectives).

| Algorithms | Explicit policy improvement | Num of trained LLMs | Inference adaptation | Preference data | Loss |
|---|---|---|---|---|---|
| MORLHF | ✓ | $M$ | × | No | PPO |
| Rewarded Soups (Ramé et al., 2023) | × | $N$ | ✓ | No | PPO |
| MODPO (Zhou et al., 2024) | ✓ | $M$ | × | Yes | DPO |
| RiC (Yang et al., 2024c) | × | 1 | ✓ | No | SFT |
| MOC (Ours) | ✓ | 1 | ✓ | No | PPO |

Can we **control** the trade-offs in a single, once-trained LLM to meet diverse human preferences? Our answer is **yes**. This paper aims to (i) enable LLMs to generate customized responses for diverse user preferences and (ii) achieve this goal with a once-trained model. To this end, we introduce a novel algorithm, Multi-Objective Control (MOC), which leverages a carefully designed multi-objective optimization (MOO) algorithm. MOC requires training only once, incorporates explicit policy improvement, and does not rely on human preference data. Moreover, its training cost is comparable to single-objective RLHF (Ouyang et al., 2022). We further improve the computational efficiency of MOC by integrating LoRA (Hu et al., 2022), making it feasible to fine-tune a 7B-parameter model on a single A6000 GPU.

We first formulate multi-objective controllability as an MOO problem with preference vector constraints, inspired by recent advancements in MOO (Désidéri, 2009; Sener & Koltun, 2018; Xiao et al., 2023). This formulation presents two primary challenges. The first is identifying the target to be controlled. Existing MOO methods typically focus on optimizing different loss functions (Liu et al., 2021; 2023) or linearized utility functions (Yang et al., 2019), which do not effectively capture the quality or behavior of LLMs. In contrast, MOC selects the reward signal as the control target, enabling direct manipulation of the model's behavior. The second challenge is to solve this optimization problem within feasible computational limits. Our formulated optimization problem involves complex trade-offs among multiple objectives under different preference constraints. To address this, we relax the problem into a new form of MOO, where the preference constraint is treated as an additional objective. MOC scalarizes the objectives with dynamic weighting in different steps, ensuring that the computational cost is comparable to the widely used single-objective RLHF (Ouyang et al., 2022). Table 1 presents a detailed comparison of MOC and baseline methods.

In extensive experiments, MOC consistently outperforms baseline methods (Ouyang et al., 2022; Ramé et al., 2023; Yang et al., 2024c) across multiple tasks. It demonstrates strong performance in three key areas: (i) controllability, as it effectively aligns model behavior with diverse preference vectors and ensures a clear monotonic relationship between input preferences and outcomes; (ii) quality of solution set, measured by the hyper-volume metric, where MOC achieves a superior Pareto front while maintaining a diverse set of solutions; and (iii) generalization, as it robustly handles unseen preferences while preserving both the alignment quality and diversity. Compared to baseline methods, MOC offers a more efficient and flexible approach to personalizing LLMs, managing different trade-offs among multiple objectives within a single model and seamlessly adapting to new preferences. These results demonstrate MOC's suitability for applications that require scalable and customizable personalization.

Our contributions are as follows: (i) We introduce the MOC algorithm, which requires comparable computation as single-objective RLHF and finetunes LLMs only once to accommodate diverse user preferences; (ii) We empirically demonstrate MOC's superior performance in terms of controllability, solution quality, and generalization, including its ability to generalize to unseen user preferences.

## 2  Multi-Objective Controllable Language Models

Modern LLM alignment is commonly implemented via RLHF (Ouyang et al., 2022): a reward model is first fit from preference pairs by maximizing

$$\mathcal{L}_{\mathrm{RM}} = \mathbb{E}_{(x,y^w,y^l)\sim\mathcal{D}}\Big[\log\sigma\big(r(x,y^w) - r(x,y^l)\big)\Big],$$

and the policy is then optimized against this learned reward, typically with PPO (Schulman et al., 2017),

$$\arg\max_{\pi(\cdot\mid x;\theta)}\mathbb{E}_{x\sim\mathcal{D},\,y\sim\pi(\cdot\mid x)}\left[r(x,y) - \beta\log\frac{\pi(y\mid x;\theta)}{\pi_{\mathrm{old}}(y\mid x)}\right]. \tag{1}$$

While effective, this pipeline implicitly commits to a *single*, developer-chosen average preference. The resulting model is well-aligned but not easily *controllable*: at inference it fails to natively adapt when users prioritize different objectives (e.g., helpfulness vs. conciseness, safety vs. creativity). We thus ask whether a model trained *once* can be steered to different trade-offs on demand. Our answer is **yes**: we aim to (i) support a spectrum of user preferences, and (ii) achieve this within one training run.

**Controllability vs. Alignment.** We use the term *controllability* to mean the model systematically varies its behavior in response to user-specified preferences, producing outputs consistent with those preferences. By contrast, *alignment* refers to optimizing toward a fixed, global preference unchanged at inference.

### 2.1  Problem Formulation

To represent user preferences, we define a preference vector $\mathbf{p} = [p_1, p_2, \cdots, p_N]$ such that $\sum_{i=1}^{N} p_i = 1$ and $p_i \geq 0$, where each element in $\mathbf{p}$ reflects the importance of a specific objective. Inspired by recent work on multi-objective learning (Xu et al., 2020; Ma et al., 2020; Yang et al., 2022), we use this preference vector to regulate the model's output in the objective space. Given a distance or divergence metric $\Phi$ and a constraint threshold $\phi$, we require the policy's deviation from the preference vectors $\{\mathbf{p}^i\}_{i=1}^{M}$ to be upper bounded by some constant $\phi$. The training of a controllable LLM is thus formulated as the following constrained optimization problem:

$$\begin{aligned}
&\max_{\theta}\mathbf{J}(\pi(\cdot;\theta,\mathbf{p})) \overset{\mathrm{def}}{=} \max_{\theta}\big(J^1(\pi(\cdot;\theta,\mathbf{p})), J^2(\pi(\cdot;\theta,\mathbf{p})), \cdots, J^N(\pi(\cdot;\theta,\mathbf{p}))\big)^\top, \\
&\text{s.t. } \Phi\Big(\pi(\cdot;\theta,\mathbf{p}) \,\Big\|\, \mathbf{p}\Big) \leq \phi,\ \forall \mathbf{p} \in \{\mathbf{p}^1, \mathbf{p}^2, \cdots, \mathbf{p}^M\},
\end{aligned} \tag{2}$$

where $J^i$ denotes the RLHF objective associated with reward $R^i$. Equation (2) ensures LLMs align with a set of preferences. The LLM is a policy $\pi$ parameterized by $\theta$ and takes a preference vector $\mathbf{p}$ as an input condition. In addition, $\Phi$ is a distance or divergence metric between the policy $\pi$ and the preference vector $\mathbf{p}$. In this work, we will later instantiate $\Phi$ as a mean squared error (MSE) between the expected reward vector and the preference vector (see Equation (3)). In this paper, the objective $J^i$ is typically selected as a PPO loss (Schulman et al., 2017; Ouyang et al., 2022), unless stated otherwise.

Traditional methods for solving constrained optimization problems, such as the Lagrangian method, are inefficient for handling the complexity of Equation (2) due to multiple constraints, diverse preferences, and the high dimensionality of LLM parameters. This limitation renders developing new solutions imperative.

### 2.2  What Should the Preference Vector Align With?

Existing multi-objective learning methods (Yang et al., 2019; Liu et al., 2023; 2021) typically focus on balancing multiple loss functions. However, RL loss does not reflect the agent's true performance and is

therefore not suitable as the target for controllability. In contrast, the value function or episodic return provides a reliable performance measure. In the context of RLHF for LLMs, the reward is evaluated by a reward model, which serves as the episodic return. Therefore, we choose a multi-dimensional reward signal as the primary target for control. To maintain simplicity, we select MSE as the similarity metric between the reward signal and the preference vector. Formally, the constraint in Equation (2) is specified as

$$\Phi\Big(\pi(\cdot;\theta,\mathbf{p}^i)\|\mathbf{p}^i\Big) \stackrel{\text{def}}{=} MSE\Big(\mathbb{E}_{x\sim\mathcal{D}}\mathbf{R}(x,y),\mathbf{p}^i\Big) \le \phi, \ \forall i \in \{1,2,\cdots,M\}, \tag{3}$$

where $x$ represents the prompt/query, $y \sim \pi(x;\theta,\mathbf{p})$ is LLM-generated response, and $\mathcal{D}$ is the prompt dataset. The reward vector $\mathbf{R}(x,y) = (R^1(x,y), R^2(x,y), \cdots, R^N(x,y))$ is associated with the $N$ optimization objectives $\{J^i\}_{i=1}^N$. The sampled response $y$ depends on the policy parameters $\theta$, enabling the optimization of $\mathbf{J}(\pi)$ with respect to $\theta$ via standard RLHF. Equation (3) enforces that the reward vector $\mathbf{R}(x,y)$ aligns closely with the preference vector $\mathbf{p}$. In other contexts such as typical RL settings, the value function can be the target of control. Further details are provided in Appendix E.

**Re-labeling the prompt.** In MOC, the policy $\pi$ takes an additional condition: the user's preference vector $\mathbf{p} = [p_1, p_2, \cdots, p_N]$. To accommodate this, we modify the original prompt by prepending the preference vector to it, as follows:

$$\text{Re-labeled prompt} = \langle\text{R1}\rangle p_1 \langle\text{R2}\rangle p_2 \cdots \langle\text{RN}\rangle p_N \{\text{prompt}\}. \tag{4}$$

## 2.3 Multi-Objective Controllability

To solve the multi-objective learning problem with inequality constraints, we introduce our MOC algorithm, which builds on recent advances of multi-objective learning (Désidéri, 2009; Sener & Koltun, 2018). MOC simultaneously optimizes all the objectives while maximizing the alignment between the objective value vector and the preference vector. For simplicity, we optimize the following similarity objective:

$$\max_\theta J^\Phi \stackrel{\text{def}}{=} \max_\theta -ReLU\Big(MSE\big(\mathbb{E}_{x\sim\mathcal{D}, y\sim\pi(x;\theta,\mathbf{p})}\mathbf{R}(x,y),\mathbf{p}\big) - \phi\Big), \ \forall\mathbf{p}\in\{\mathbf{p}^1,\mathbf{p}^2,\cdots,\mathbf{p}^M\}, \tag{5}$$

where $ReLU(x) = \max(x,0)$ penalizes constraint violations when the error exceeds the threshold $\phi$. This ensures that optimization respects the trade-offs between rewards and preferences. For brevity, we denote the expectation over the data and policy distribution $\mathbb{E}_{x\sim\mathcal{D},y\sim\pi(x;\theta,\mathbf{p})}$ simply as $\mathbb{E}$. For notational simplicity, in the rest of this paper we write $\mathbf{p}$ to denote an arbitrary element of the finite preference set $\{\mathbf{p}^i\}_{i=1}^M$ (i.e., we omit the superscript $i$ when it is clear from context). This is purely a notational convention: $\mathbf{p}$ always refers to an element of $\{\mathbf{p}^i\}_{i=1}^M$. When a statement is required to hold for all preference vectors, we will write it explicitly as $\forall\mathbf{p}\in\{\mathbf{p}^i\}_{i=1}^M$. In practice, we sample preference vectors from this set during training. The gradient of Equation (5) can be approximated as

$$\nabla_\theta\text{ReLU}(\text{MSE}(\mathbb{E}\mathbf{R}(x,y),\mathbf{p})-\phi) = \mathbf{1}_{\text{MSE}(\mathbb{E}\mathbf{R}(x,y),\mathbf{p})-\phi>0}\sum_{k=1}^N[(\mathbb{E}R^k - p_k)\nabla_\theta\mathbb{E}R^k(x,y)], \tag{6}$$

where $\mathbf{1}_{(\cdot)}$ is the indicator function, $R^k$ represents the $k$-th entry of $\mathbf{R}$, $p_k$ means the $k$-th entry of preference vector $\mathbf{p}$. The term $\mathbb{E}_{x\sim\mathcal{D},y\sim\pi(x;\theta,\mathbf{p})}R^k(x,y)$ aims at maximizing the corresponding reward, which is also the goal of the gradient of the PPO loss. Thus, one could use the PPO objective $\nabla_\theta J^k(\pi(\cdot;\theta,\mathbf{p}))$ to compute $\mathbb{E}_{x\sim\mathcal{D},y\sim\pi(x;\theta,\mathbf{p})}R^k(x,y)$.

Solving the original optimization problem in Equation (2) is computationally challenging due to the involvement of $N$ objectives and $M$ preferences. Thus, we relax and reformulate it as

$$\max_\theta\widehat{\mathbf{J}}(\pi(\cdot;\theta,\mathbf{p})) \stackrel{\text{def}}{=} \max_\theta\Big(\mathbf{p}^\top\mathbf{J}(\pi(\cdot;\theta,\mathbf{p})), -\text{ReLU}\Big(\text{MSE}\big(\mathbb{E}_{x\sim\mathcal{D}}\mathbf{R}(x,y),\mathbf{p}\big)-\phi\Big)\Big)^\top, \tag{7}$$

where $\mathbf{J}(\pi(\cdot;\theta,\mathbf{p}))$ is defined in Equation (2). Equation (7) is a relaxation of Equation (2), not a strictly equivalent reformulation. The second objective is a hinge penalty on the constraint residual $\text{MSE}(\mathbb{E}\mathbf{R},\mathbf{p})-\phi$,

which is maximized at 0. This reformulation offers two significant advantages: (i) It significantly reduces optimization complexity by transforming the original $N$-objective optimization into a bi-objective optimization; (ii) It retains control over the preference vectors in the newly formulated optimization problem. Scalarization simplifies the problem even further:

$$\max_\theta \left\{ c^{(1)} \mathbf{p}^\top \mathbf{J}(\pi(\cdot;\theta,\mathbf{p})) - c^{(2)} \mathrm{ReLU}\Big( \mathrm{MSE}(\mathbb{E}_{x\sim\mathcal{D}}\mathbf{R}(x,y),\mathbf{p}) - \phi \Big) \Big| \sum_{i=1}^2 c^{(i)} = 1, \ c^{(i)} \geq 0 \right\}, \tag{8}$$

where $c^{(i)}$ is an $i$-objective related coefficient, determined by solving a min-norm problem

$$\min_{c^{(1)},c^{(2)}} \left\{ \left\| c^{(1)} \mathbf{p}^\top \nabla_\theta \mathbf{J}(\pi(\cdot;\theta,\mathbf{p})) - c^{(2)} \nabla_\theta \mathrm{ReLU}\Big( \mathrm{MSE}(\mathbb{E}_{x\sim\mathcal{D}}\mathbf{R}(x,y),\mathbf{p}) - \phi \Big) \right\|_2^2 \Big| \sum_{i=1}^2 c^{(i)} = 1, \ c^{(i)} \geq 0 \right\}. \tag{9}$$

As demonstrated by Désidéri (2009), either: (i) The solution to this min-norm problem is zero, meaning it satisfies the KKT conditions; or (ii) it yields a gradient direction that improves all objectives.

### 2.4 Scalable Multi-Objective Control of LLMs with Surrogate Objectives

However, in the context of LLMs, directly solving this optimization remains computationally intractable due to: (i) the need to backpropagate $N + 1$ times to compute the gradient for each objective; and (ii) the prohibitive computational expense of solving the min-norm problem in the gradient space for LLM parameters. To overcome this computational burden, we introduce a more computationally efficient surrogate, which is an upper bound to the original objective, circumventing the need for costly backpropagation operations.

**Theorem 1.** *Let $z(\theta) = \frac{\pi(y|x;\theta)}{\pi_{old}(y|x)}$ denote the probability ratio of PPO objective (Equation (1)), and let $\epsilon$ be the clipping hyper-parameter as defined in PPO (Schulman et al., 2017). The upper bound of objective in Equation (9) is given by*

$$\left\| c^{(1)} \sum_{j=1}^N p_j I(\hat{A}_j) - c^{(2)} \mathbf{1}_{MSE(\mathbb{E}_{x\sim\mathcal{D}}\mathbf{R}(x,y),\mathbf{p})-\phi>0} \sum_{j=1}^N (R^j - p_j) I(\hat{A}_j) \right\|_2^2 \times \left\| \nabla_\theta \pi(\cdot;\theta,\mathbf{p}) \right\|_2^2, \tag{10}$$

*where*

$$I(A) = \begin{cases} 0, & if \ (A > 0 \ and \ z > 1 + \epsilon) \\ & \quad or \ (A < 0 \ and \ z < 1 - \epsilon), \\ A, & if \ (A > 0 \ and \ z \leq 1 + \epsilon) \\ & \quad or \ (A < 0 \ and \ z \geq 1 - \epsilon), \end{cases} \tag{11}$$

$$\sum_{i=1}^2 c^{(i)} = 1, \quad c^{(i)} \geq 0 \quad \forall i, \tag{12}$$

*where $\hat{A}_j$ denotes the advantage estimate.*

The proof is deferred to Appendix A. Theorem 1 provides an upper bound on Equation (9), which yields two key advantages: (i) Both $I(\hat{A}_i)$ and $\mathbf{1}_{MSE(\mathbb{E}_{x\sim\mathcal{D}}\mathbf{R}(x,y),\mathbf{p})-\phi>0} \sum_{j=1}^N (R^j - p_j) I(\hat{A}_j)$ can be efficiently computed without any additional expensive back-propagation; (ii) $\nabla_\theta \pi(\cdot;\theta,\mathbf{p})$ is no longer required by the min-norm problem as it does not depend on $c^{(i)}$. Therefore, we achieve the following computationally efficient surrogate problem of optimizing $c^{(1)}$ and $c^{(2)}$:

$$\min_{c^{(1)},c^{(2)}} \left\{ \left\| c^{(1)} \sum_{j=1}^N p_j I(\hat{A}_j) - c^{(2)} \mathbf{1}_{MSE(\mathbb{E}_{x\sim\mathcal{D}}\mathbf{R}(x,y),\mathbf{p})-\phi>0} \sum_{j=1}^N (R^j - p_j) I(\hat{A}_j) \right\|_2^2 \Big| \sum_{i=1}^2 c^{(i)} = 1, \ c^{(i)} \geq 0, \ \forall i \right\}. \tag{13}$$

Compared to the intractable original optimization in Equation (9), the surrogate optimization problem in Equation (13) offers the following advantages: (i) **Computational efficiency**: The term $I(\hat{A}_i)$ can be

computed through a simple forward pass in a language model without requiring gradient calculations; (ii) **Solution efficiency**: Note that the objective function is a quadratic function of the variables $c^{(i)}$. The general min-norm problem is solvable by the existing Frank-Wolfe algorithm (Jaggi, 2013), a well-established convex optimization method. Equation (13) has a closed-form solution (Sener & Koltun, 2018) because Equation (13) only involves two terms.

As a result, the multi-objective learning problem in Equation (8) can be solved by iterating two steps: (i) Solving the min-norm problem in Equation (13) to achieve the dynamic weights $\{c^{(i)}\}_{i=1}^2$, and (ii) Optimizing scalarized objective in Equation (8) with the $\{c^{(i)}\}_{i=1}^2$. Finally, by integrating PPO's advantage function $A$ into Equation (13), our MOC algorithm can train a policy taking any preference vector to control the multi-objective alignment. This algorithm is summarized in Appendix D.

**Advantages of MOC**. We highlight the benefits of our approach: i) **Flexible preference handling**: MOC can accommodate a wide range of preferences through a single training process. Its design enables adaptation to diverse preferences without repeated training. ii) **Computational Efficiency**: By introducing the surrogate objective in Equation (13), MOC significantly reduces computational costs, making its cost comparable to widely used single-objective RLHF.

## 2.5 An Illustrative Example

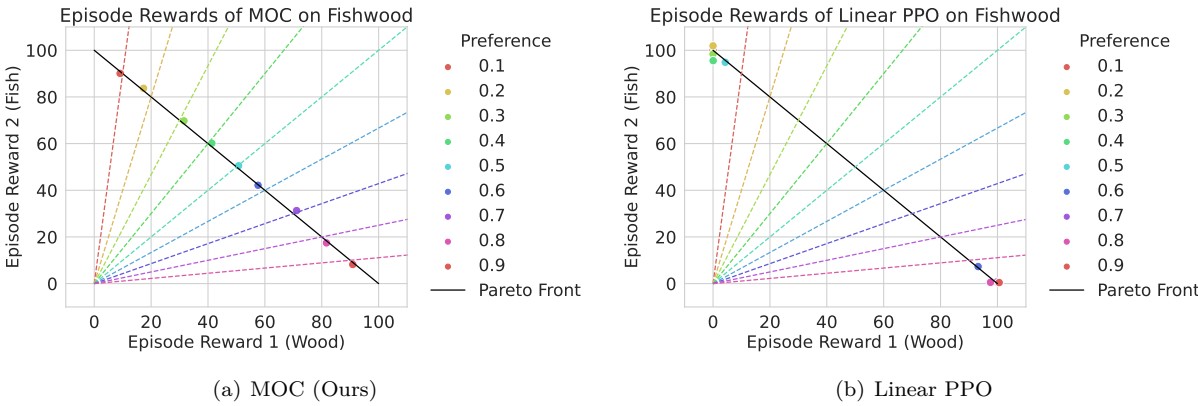

(a) MOC (Ours)               (b) Linear PPO

Figure 1: Solutions of MOC and Linear PPO on fishwood task and the Pareto front (line in black). MOC demonstrates advantages in both MOO (solutions lie on the Pareto front) and multi-objective control (its solutions align closely with their corresponding preference vectors, shown as the colored dashed rays). The single model trained by MOC can handle diverse preference vectors. In contrast, Linear PPO optimizes a linear scalarization of the objectives and fails to follow the preference vectors, with solutions dominated by one objective. The legend "Preference" indicates the specific weight value assigned to Reward 1 (Wood). Linear PPO is implemented by optimizing PPO w.r.t. a scalarized reward $R_{\text{lin}} = w\,R^{\text{wood}} + (1-w)\,R^{\text{fish}}$ (with $w \in [0, 1]$). In our experiments, we train Linear PPO runs with weights $[0.1, 0.9], [0.2, 0.8], \cdots, [0.8, 0.2], [0.9, 0.1]$.

To demonstrate the capability of our proposed MOC algorithm, we perform an illustrative experiment on the fishwood task (Felten et al., 2023), where the agent controls a fisherman who can fish or gather wood, receiving corresponding rewards upon completion of each task. The rewards have two dimensions: one for gathering wood and one for fishing. Each collected wood or fished item increases the respective reward by 1. Detailed experimental setup can be found in Appendix G. The results are reported in Figure 1. MOC aims for (i) MOO: The solutions should reach the Pareto front, meaning the points should be close to the black solid line. (ii) Multi-objective control: The solutions should align closely with their respective preference vectors, indicated by the dashed lines.

The results demonstrate that MOC achieves both goals. (i) Its solutions lie on the Pareto front, demonstrating successful optimization, and (ii) its solutions are close to the preference vectors, confirming effective multi-objective control. Notably, MOC generalizes to diverse preference vectors by training only **one** model. In contrast, the Linear PPO method, which optimizes a linear scalarization of the objectives, struggles to

consistently follow different preference vectors. In Linear PPO's results, one objective often dominates the other in the Pareto sense, a well-known phenomenon in convex optimization (see section 4.7 of Boyd & Vandenberghe (2004)).

## 3 Experiments

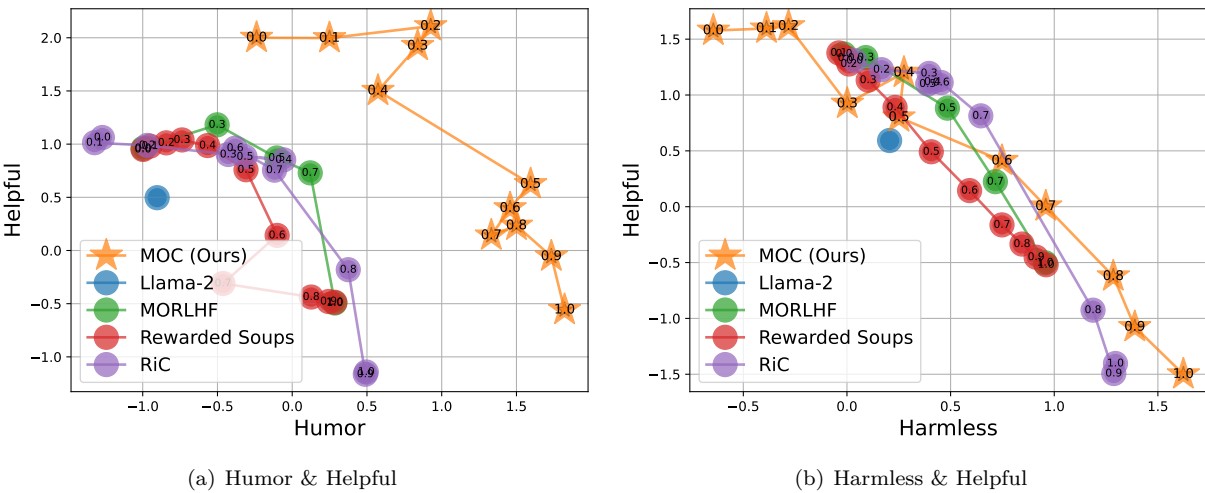

(a) Humor & Helpful          (b) Harmless & Helpful

Figure 2: **Controllability comparison on the Pareto front**. MOC demonstrates superior controllability, indicated by the consistent ranking of solutions on their preference weights and the achieved reward values. In comparison, the baselines exhibit less stable behavior and weaker alignment with the specified preferences. MOC also achieves higher quality solutions, particularly in the Humor & Helpful alignment. Our MOC method achieves the best overall performance, supported by these results and the findings in Tables 2 to 4. Each point represents the reward achieved across multiple instances, each with a different input preference vector. Each point's preference weight for the x-axis reward is the numerical label on its marker.

In this section, we present a comprehensive evaluation of the proposed MOC method.

### 3.1 Experimental Setup

**Implementation.** Our implementation is based on the open-source TRL package (von Werra et al., 2020). For the language model, we adopt models from the Llama series (Touvron et al., 2023; Dubey et al., 2024), which are widely used in RLHF studies. We use the prompts from Helpful Assistant dataset (Bai et al., 2022), which provides data for two sets of objectives: {"humor", "helpful"} and {"harmless", "helpful"}. MOC is trained with a set of predefined preference vectors: {[0.0, 1], [0.1, 0.9], · · · , [0.9, 0.1], [1, 0.0]}. The training process was performed on a desktop with an Intel i9-14900K CPU and an NVIDIA RTX A6000 GPU. MOC is trained by LoRA (Hu et al., 2022) with a rank of 64. The language model is loaded in 8-bit due to the computational constraints. Additional experimental details are provided in Appendix H. While human evaluation would provide stronger evidence, we follow standard RLHF practice and use reward-model-based rankings as a proxy for human preferences, since reward models are trained from human preference pairs.

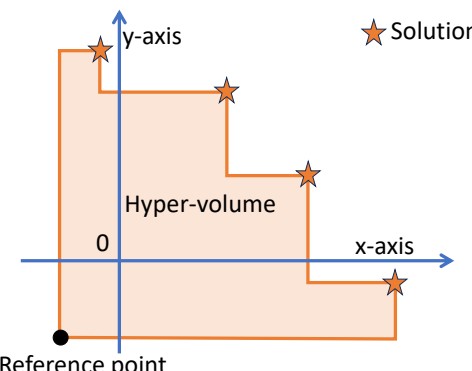

Figure 3: Illustration of the hyper-volume concept. The hyper-volume measures the size of the objective space dominated by a set of solutions in multi-objective optimization. Larger hyper-volumes indicate better convergence and diversity of the Pareto front.

**Baselines.** We compare MOC against three baselines, including (i) The standard *MORLHF*: a multi-objective RLHF method that scalarizes the multi-objective problem into a single objective by combining reward signals with fixed preference weights (in our experiments, the fixed preference weights are $\{[0.0, 1.0], [0.3, 0.7], [0.5, 0.5], [0.7, 0.3], [1.0, 0.0]\}$). MORLHF does not condition the policy on the preference vector. These weights are only used to scalarize multiple reward-model scores into a single scalar reward, which is then passed to PPO. (ii) *Rewarded Soups* (Ramé et al., 2023): Combines the model weights of $N$ separately trained models using the PPO algorithm, each optimized for a specific reward function; (iii) *RiC* (Yang et al., 2024c) learns a mapping from preference vectors to reward signals and then conditions generation by injecting the corresponding reward signals into the prompt (i.e., reward-as-context prompt conditioning). RiC is trained with SFT. Rewarded Soups and RiC use the same set of preference vectors as MOC. The behavior of the base Llama-2 model is included for comparative analysis. Further discussion of RiC and other related work is provided in Appendix F. More baselines are included in Appendix K.

**Metrics.** It is crucial to emphasize that in the context of controllability, higher rewards do **NOT** always equate to better outcomes. Our goal is to ensure that the model's outputs align with the personalized expectations of users, rather than merely maximizing reward scores. Therefore, to evaluate the controllability comprehensively, we employ a multi-faceted approach.

The evaluation focuses on four key aspects, namely: (i) The quality of solution set, measured using hypervolumes (as illustrated in Figure 3); (ii) Control with preference vectors, assessed by computing the correlation between the model's behavior and the given preferences; (iii) Diversity of solutions, evaluated by computing the mean pairwise distance of the solutions; and (iv) Generalization capabilities to unseen preference vectors. These metrics collectively provide a robust principle to evaluate the model's ability to optimize competing objectives while adhering to user-defined preferences. Additionally, we present case studies to provide qualitative insights into the controllability of MOC with user-specified preferences.

## 3.2 Main Results

Figure 2 illustrates the results for two pairs of reward models, with coordinates representing the average rewards corresponding to different preference vectors.

Our evaluation reveals two primary findings: (i) **Controllability**: MOC demonstrates superior controllability compared to the baselines. This is evident in how consistently the model's behavior aligns with the rank order prescribed by the preference vectors, maintaining a clear monotonic relationship between given preferences and corresponding rewards. In contrast, MORLHF, Rewarded Soups, and RiC exhibit less consistent behavior under corresponding preference settings; (ii) **Solution quality**: MOC dominates the baselines in terms of overall Pareto front quality, as confirmed by the quantitative results below.

Table 2: Controllability comparison using Kendall's tau correlation (higher is better), with corresponding $p$-values in parentheses (smaller is better), measuring the consistency between input preferences and output rewards. MOC significantly outperforms all the baselines.

| Setting | MOC (Ours) | RiC | MORLHF | Rewarded Soups |
|---|---|---|---|---|
| Humor-helpful | 1.00 ($5.0 \times 10^{-8}$) | 0.78 ($3.3 \times 10^{-4}$) | 1.00 ($1.7 \times 10^{-2}$) | 1.00 ($5.0 \times 10^{-8}$) |
| Harmless-helpful | 1.00 ($5.0 \times 10^{-8}$) | 0.91 ($3.0 \times 10^{-5}$) | 1.00 ($1.7 \times 10^{-2}$) | 0.96 ($5.5 \times 10^{-7}$) |
| Average | 1.00 | 0.85 | 1.00 | 0.98 |

**Alignment with preferences.** To assess how well different algorithms align with the prescribed preference vectors and model behavior, we adopt *Kendall's tau* rank correlation (Kendall, 1938) as the evaluation metric. Kendall's tau quantifies the degree to which the relative ordering of outputs is consistent with the rank order determined by the preference vectors, capturing the model's ability to maintain rank-preserving relationships. A higher Kendall's tau indicates better alignment between the model's output ordering and the target preference ordering. Concretely, we compute an angular projection score for each output based on the angle (computed via arctan) of its reward vector relative to a fixed reference point, rank outputs by this

score, and then compute Kendall's tau against the preference-induced ranking (see Appendix I for details). The results are shown in Table 2. Across two evaluation settings, MOC achieves the highest Kendall's tau, demonstrating its superior capability to align model behavior with user preferences and accurately reflect human preference rankings.

**Quality of solutions.** We use the hyper-volume indicator, a standard metric in MOO, to measure the quality of solutions. Hyper-volume captures both convergence to the Pareto front and the diversity of the solutions in the objective space. Table 3 shows that MOC significantly outperforms all baselines. For instance, in the Humor-Helpful setting, MOC achieves a hyper-volume of 14.176, compared to 6.692 by RiC; similar trends are observed in the Harmless-Helpful setting. These results indicate that MOC exhibits superior convergence to the Pareto front and maintains a more diverse set of solutions, ensuring that it explores a broader range of trade-offs between objectives.

Table 3: Hyper-volume (higher is better) comparison of different methods, measuring the volume of solutions dominated by each method's solution set. MOC outperforms all baselines, achieving higher solution diversity and quality. The best score is marked with the `blue` color box.

| Setting | MOC (Ours) | RiC | MORLHF | Rewarded Soups |
|---|---|---|---|---|
| Humor-helpful | 14.176 | 6.692 | 6.769 | 6.100 |
| Harmless-helpful | 10.220 | 9.257 | 9.047 | 8.905 |
| Average | 12.198 | 7.974 | 7.908 | 7.502 |

Table 4: Comparison of solution set pair-wise distances (measuring diversity, higher is better) across methods.

| Setting | MOC (Ours) | RiC | MORLHF | Rewarded Soups |
|---|---|---|---|---|
| Humor-helpful | 1.439 | 1.260 | 1.057 | 1.005 |
| Harmless-helpful | 1.600 | 1.363 | 1.110 | 1.015 |
| Average | 1.520 | 1.312 | 1.084 | 1.010 |

**Diversity of solutions.** We measure the diversity of solutions by computing the mean pairwise distance (MPD) of the solution set in the objective space. MPD directly quantifies how spread out the solutions are, making it particularly suitable for our setting. A higher MPD indicates greater behavioral diversity. Table 4 shows that MOC consistently achieves the highest MPD values, outperforming all baselines. For example, in the Harmless-Helpful setting, MOC obtains an MPD value of 1.600, while RiC obtains 1.363. This result aligns with the observation in Figure 2, where the solutions produced by RiC tend to cluster more closely, resulting in less diverse behavior.

### 3.3 Generalization to Unseen User Preference

In this work, we define "unseen preferences" as valid preference vectors (where $\sum p_i = 1$) sampled from the continuous probability simplex that were not present in the training set. Since our training set includes the boundary vertices of the preference simplex, evaluating on these unseen preference vectors strictly assesses the model's capability for continuous interpolation across the Pareto front, ensuring the model has learned a smooth manifold rather than discrete memorization. We evaluate the ability of our model to generalize to unseen preference vectors not included in the training process. Although MOC is initially trained on a predefined set of preference vectors, the goal is to determine if it can handle new, untrained preferences effectively. To test this, we uniformly sampled four sets of unseen preference vectors and provided them as inputs to the trained model for inference. The results, as depicted in Figure 4, confirm that the model maintains strong performance across all tested scenarios, without any obvious degradation in its behavior. Check Appendix J for more details and qualitative results.

The results highlight several key advantages of the model trained by MOC: i) The model's performance does not degrade when presented with unseen preferences. ii) The model's behavior still adheres to the input

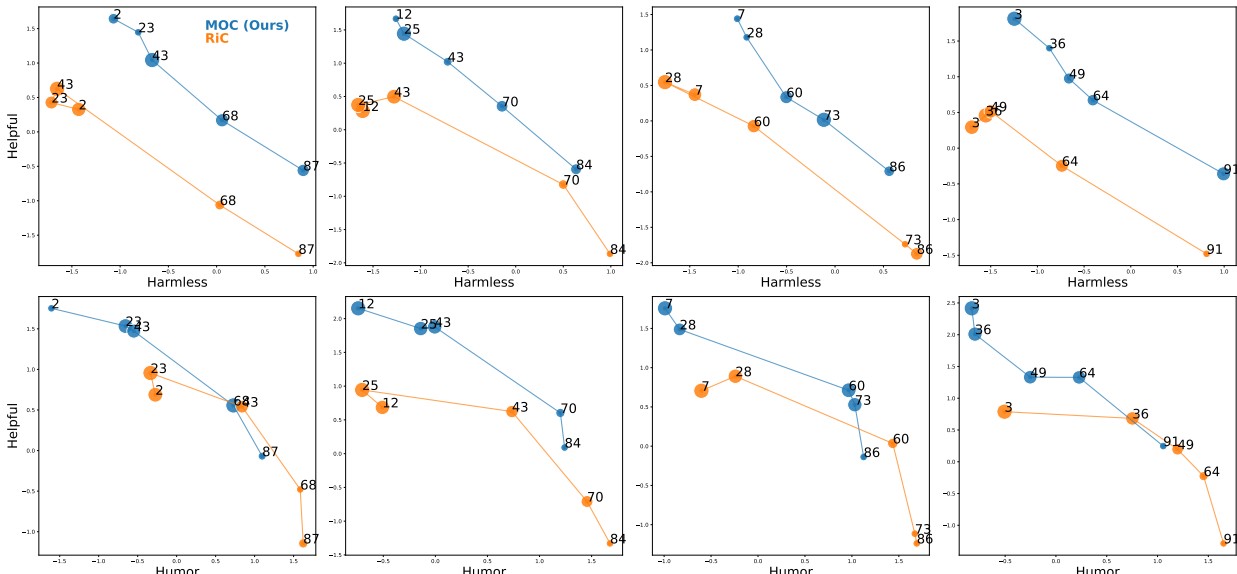

Figure 4: Generalization to unseen preference vectors held out from the training. MOC and RiC-trained LLMs are compared on four random sets of unseen preference vectors. Each column corresponds to a different set of unseen preference vectors, and each row represents a different pair of reward settings. MOC solutions dominate the RiC solutions in most cases. MOC's rewards align with the new preference vectors and the outputs under different preferences are diverse in the reward space. This suggests MOC generalizes to unseen preferences and achieves diverse trade-offs on the Pareto front. The size of each point indicates the standard deviation in rewards. The numerical labels indicate the preference weights (multiplied by 100) for the reward on the x-axis, enhancing visual clarity.

preference vector, maintaining alignment between behavior ranking (represented by rewards) and preferences. iii) The model demonstrates sufficient diversity in its behavior, distributing its rewards across a broad range of outcomes rather than concentrating on a narrow region of the objective space. These results suggest that MOC can successfully accommodate a diverse range of trade-offs dictated by new preferences, even when they significantly differ from those encountered during training.

### 3.4 Additional Experiments

To further demonstrate MOC's capabilities, we present a series of experiments highlighting its advantages across various settings. These include: (1) its ability to generalize to untrained preference vectors (Appendix J), (2) its performance across different model types and sizes, compared with more baselines (Appendix K), (3) its adaptability to various datasets and reward models (Appendix L), (4) its effectiveness in handling more objectives (Appendices M and N), (5) ablation study (Appendix O), and (6) a case study (Appendix P).

### 3.5 Discussion

The experiments reveal four key advantages of MOC. (i) **Solution Quality**: MOC achieves the highest solution quality, evidenced by hyper-volume, reflecting convergence and diversity. (ii) **Controllability**: MOC demonstrates superior controllability, ensuring consistent alignment with user preferences across diverse objective trade-offs. (iii) **Solution Diversity**: MOC outperforms baselines, confirming its robustness in capturing user preferences. (iv) **Generalization**: MOC's ability to generalize to unseen preferences highlights its potential for real-world applications where new preferences may emerge. These advantages demonstrate MOC offers a powerful and flexible approach for multi-objective controllable language models, outperforming baselines in controllability and diversity while maintaining computational efficiency.

## 4 Related Work

The alignment of LLMs with human values is a central challenge (Ouyang et al., 2022; Bai et al., 2022), and moving beyond a single, developer-defined preference to accommodate diverse, multi-objective user needs is an open problem. We first review foundational MOO and then situate MOC within the current landscape of multi-objective control methods for LLMs.

### 4.1 From Multi-Objective Optimization to Controllable LLMs

The principles of MOO, which aim to find a set of Pareto-optimal solutions for competing objectives, are well-established (Désidéri, 2009; Sener & Koltun, 2018). However, applying these principles to large-scale LLMs presents unique challenges. Early MOO work in machine learning often focused on optimizing multiple loss functions simultaneously (Liu et al., 2021; 2023), a paradigm that does not directly map to controlling the fine-grained, semantic behaviors of LLMs.

Closer to our domain, some methods rely on a linear scalarization of utility or reward functions (Yang et al., 2019), a technique also employed by standard MORLHF. While simple, this approach is theoretically limited and often fails to identify solutions in non-convex regions of the Pareto front (Boyd & Vandenberghe, 2004). Other algorithms trace the full Pareto front (Mahapatra & Rajan, 2021; Zhang et al., 2024), but their high computational complexity makes them intractable for finetuning billion-parameter LLMs.

MOC addresses this challenge. Our work differs from these foundational methods in two ways: (i) we directly manipulate model behavior in the more meaningful reward space rather than loss space, and (ii) we introduce a novel surrogate objective (Theorem 1) that makes principled MOO computationally efficient and comparable to standard single-objective RLHF, thus overcoming the scalability bottleneck of prior work.

### 4.2 Multi-Objective Control of LLMs

Recent years have seen several methods specifically for multi-objective LLM control, which can be broadly categorized by their approach.

**Training a Single Steerable Policy.** This paradigm, which MOC belongs to, aims to train one versatile model that can be steered at inference time. The most related works use a numerical vector to specify preferences. Wang et al. (2024) provides a foundational RL framework for this but, like MORLHF, relies on linear reward scalarization. Guo et al. (2024) presents a powerful, RL-free alternative by extending DPO to handle multiple objectives, making it a key intellectual parallel to our work. RiC (Yang et al., 2024c) conditions the model on reward values via prompt engineering and uses rejection sampling. While this achieves a form of control, it lacks an explicit policy improvement mechanism, limiting its ability to push the Pareto frontier outwards. MOC, by integrating its MOO formulation directly into a PPO-based policy-gradient objective, explicitly optimizes for both alignment and reward maximization. In contrast to these, MOC employs a more sophisticated MOO gradient calculation, enabling a superior exploration of the Pareto front, as demonstrated by our hyper-volume and diversity metrics (Tables 3 and 4).

**Ensemble and Multi-Model Methods.** Other methods achieve diverse outputs by training or combining multiple models. Rewarded Soups (Ramé et al., 2023) interpolates the weights of separate models, each finetuned on a single reward. Similarly, MODPO (Zhou et al., 2024) trains M distinct models for M preferences. These approaches are computationally expensive and storage-intensive, directly contrasting with MOC's "one model for all" design. MOC achieves superior results with the efficiency of training a single model.

### 4.3 Alternative Control Interfaces and Methodologies

**Linguistic and Implicit Control.** Some works explore more user-friendly control interfaces. Nguyen et al. (2024); Yang et al. (2024a) use explicit linguistic tags (e.g., [formality: high]) for control, while MOSLIM (Zhang et al., 2025) infers preferences implicitly from the user's natural language prompt. These methods trade the precise, granular control offered by MOC's numerical vectors for enhanced interpretability

or zero-shot prompting. MOC's approach is complementary and particularly suited for applications requiring precise, backend control over model attributes like safety and factuality.

**Inference-Time Control.** Another approach is to enforce constraints during decoding. Shi et al. (2024); Son et al. (2025) pioneer methods to provide formal guarantees on objective trade-offs at inference time. This technique applies to any LLM but incurs latency and can degrade text quality if constraints are too severe. MOC, as a training-time method, internalizes these trade-offs, enabling fast, coherent, and controlled generation without specialized decoding, making it practical for deployment.

MOC provides a unique and powerful framework. It is a single, efficient, and steerable model that: (i) does not require training multiple models, unlike Rewarded Soups or MODPO; (ii) does not rely on preference datasets, unlike CPO (Guo et al., 2024); (iii) maintains explicit policy improvement, unlike RiC; and (iv) generalizes to unseen preference vectors, which is a primary design goal.

## 5  Conclusion

In this paper, we introduced MOC, a novel approach that enables the personalization of LLMs by adapting to diverse user-specified preferences. MOC addresses the limitations of existing LLMs, which are typically constrained by fixed developer-specified preferences, by formulating multi-objective controllability as a multi-objective optimization problem. By introducing surrogate optimization in RLHF, MOC enables a single fine-tuning process to adapt to a wide range of user-specified trade-offs while many existing methods require training multiple separate models. Our experiments demonstrate that MOC surpasses existing baseline methods in controllability, solution quality, and generalization while maintaining exceptional computational efficiency. By optimizing over the explicit Pareto front rather than collapsed scalar values, MOC enables precise, continuous control over multi-dimensional model behaviors. This work highlights the potential of MOC to transform how LLMs interact with users, offering scalable and customizable solutions that meet diverse needs while maintaining computational feasibility. Looking forward, MOC paves the way for fully personalized systems. While our framework utilizes precise numeric vectors for optimization, it effectively serves as a foundational control layer compatible with intuitive user interfaces. Specifically, natural language instructions can be mapped to these continuous vectors through a lightweight translation module, connecting human intent with fine-grained model control. This modular design facilitates the integration of MOC into real-world applications where both interpretability and precision matter. Ultimately, MOC represents a significant step toward realizing fully personalized and human-friendly systems.

### Acknowledgments

The work of Q. H. and S. M. was supported by the Federal Ministry of Research, Technology and Space (BMFTR) under Grant 16KIS2411. The work of Y. Y. and M. P. was supported by the TKI SmartTwo project. The role of Y. Y. as the primary contributor to the algorithm design and theoretical components is explicitly acknowledged here.

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

## Appendix Table of Contents

# A    Proof of Theorem 1

**Theorem 1.** *Let $z(\theta) = \frac{\pi(y|x;\theta)}{\pi_{old}(y|x)}$ denote the probability ratio of PPO objective (Equation (1)), and let $\epsilon$ be the clipping hyper-parameter as defined in PPO (Schulman et al., 2017). The upper bound of objective in Equation (9) is given by*

$$\left\| c^{(1)} \sum_{j=1}^{N} p_j I(\hat{A}_j) - c^{(2)} \mathbf{1}_{MSE(\mathbb{E}_{x \sim \mathcal{D}} \mathbf{R}(x,y), \mathbf{p}) - \phi > 0} \sum_{j=1}^{N} (R^j - p_j) I(\hat{A}_j) \right\|_2^2 \times \left\| \nabla_\theta \pi(\cdot; \theta, \mathbf{p}) \right\|_2^2, \tag{10}$$

*where*

$$I(A) = \begin{cases} 0, & \text{if } (A > 0 \text{ and } z > 1 + \epsilon) \\ & \quad \text{or } (A < 0 \text{ and } z < 1 - \epsilon), \\ A, & \text{if } (A > 0 \text{ and } z \leq 1 + \epsilon) \\ & \quad \text{or } (A < 0 \text{ and } z \geq 1 - \epsilon), \end{cases} \tag{11}$$

$$\sum_{i=1}^{2} c^{(i)} = 1, \quad c^{(i)} \geq 0 \quad \forall i, \tag{12}$$

*where $\hat{A}_j$ denotes the advantage estimate.*

*Proof.* One can further expand Equation (9) with the PPO loss and get

$$
\begin{aligned}
& \left\| c^{(1)} \mathbf{p}^\top \nabla_\theta \mathbf{J}(\pi(\cdot; \theta, \mathbf{p})) - c^{(2)} \nabla_\theta ReLU(MSE(\mathbb{E}_{x \sim \mathcal{D}} \mathbf{R}(x,y), \mathbf{p}) - \phi) \right\|_2^2 \\
= & \left\| c^{(1)} \sum_{j=1}^{N} p_j \nabla_\theta J^j(\pi(\cdot; \theta, \mathbf{p})) - c^{(2)} \nabla_\theta ReLU(MSE(\mathbb{E}_{x \sim \mathcal{D}} \mathbf{R}(x,y), \mathbf{p}) - \phi) \right\|_2^2 \\
= & \left\| c^{(1)} \sum_{j=1}^{N} p_j \nabla_\pi J^j(\pi(\cdot; \theta, \mathbf{p})) \nabla_\theta \pi(\cdot; \theta, \mathbf{p}) - c^{(2)} \nabla_\pi ReLU(MSE(\mathbb{E}_{x \sim \mathcal{D}} \mathbf{R}(x,y), \mathbf{p}) - \phi) \nabla_\theta \pi(\cdot; \theta, \mathbf{p}) \right\|_2^2 \\
\leq & \left\| c^{(1)} \sum_{j=1}^{N} p_j \nabla_\pi J^j(\pi(\cdot; \theta, \mathbf{p})) - c^{(2)} \nabla_\pi ReLU(MSE(\mathbb{E}_{x \sim \mathcal{D}} \mathbf{R}(x,y), \mathbf{p}) - \phi) \right\|_2^2 \left\| \nabla_\theta \pi(\cdot; \theta, \mathbf{p}) \right\|_2^2 \\
= & \left\| c^{(1)} \sum_{j=1}^{N} p_j \frac{1}{\pi_{old}} I(\hat{A}_j) - c^{(2)} \mathbf{1}_{MSE(\mathbb{E}_{x \sim \mathcal{D}} \mathbf{R}(x,y), \mathbf{p}) - \phi > 0} \sum_{j=1}^{N} (R^j - p_j) \frac{1}{\pi_{old}} I(\hat{A}_j) \right\|_2^2 \left\| \nabla_\theta \pi(\cdot; \theta, \mathbf{p}) \right\|_2^2 \\
\leq & \left\| c^{(1)} \sum_{j=1}^{N} p_j I(\hat{A}_j) - c^{(2)} \mathbf{1}_{MSE(\mathbb{E}_{x \sim \mathcal{D}} \mathbf{R}(x,y), \mathbf{p}) - \phi > 0} \sum_{j=1}^{N} (R^j - p_j) I(\hat{A}_j) \right\|_2^2 \left\| \nabla_\theta \pi(\cdot; \theta, \mathbf{p}) \right\|_2^2
\end{aligned}
\tag{14}
$$

where

$$I(A) = \begin{cases} 0, & \text{if } (A > 0 \text{ and } z > (1 + \epsilon)) \\ & \quad \text{or } (A < 0 \text{ and } z < 1 - \epsilon) \\ A, & \text{if } (A > 0 \text{ and } z \leq (1 + \epsilon)) \\ & \quad \text{or } (A < 0 \text{ and } z \geq 1 - \epsilon) \end{cases},$$

$$\sum_{i=1}^{2} c^{(i)} = 1, \quad c^{(i)} \geq 0 \quad \forall i,$$

and $z = \frac{\pi}{\pi_{old}}$. The third inequality holds by Cauchy–Schwarz inequality and the fourth equation holds by integrating the PPO loss function. □

# B Pareto Optimality and MOC's Advantages

In this section, we provide a formal definition of Pareto Optimality and its relevance to policy improvement.

## Formal Definition of Pareto Optimality

**Definition 1.** Let $\pi, \pi' \in \mathcal{X}$, where $\mathcal{X}$ is the set of feasible solutions. A solution $\pi$ is said to *dominate* another solution $\pi'$ if and only if:

- $J_i(\pi) \geq J_i(\pi')$ for all $i \in \{1, 2, \ldots, N\}$, and

- $J_j(\pi) > J_j(\pi')$ for at least one $j \in \{1, 2, \ldots, N\}$.

Here, $J_i(\pi)$ denotes the value of the $i$-th objective for the solution $\pi$. The above conditions imply that $\pi$ performs at least as well as $\pi'$ in all objectives and strictly better in at least one. Solutions that are not dominated by any other are termed *non-dominated* and collectively form the *Pareto front*.

**Definition 2.** (Pareto Optimality) Let $\mathcal{X}$ denote the set of feasible solutions, and let $J : \mathcal{X} \to \mathbb{R}^N$ be a vector-valued objective function where $J(\pi) = [J_1(\pi), J_2(\pi), \ldots, J_N(\pi)]^\top$ corresponds to the objective values associated with $\pi \in \mathcal{X}$. A solution $\pi^* \in \mathcal{X}$ is *Pareto optimal* if and only if no other solution $\pi' \in \mathcal{X}$ satisfies:

$$J_i(\pi') \geq J_i(\pi^*) \quad \forall i \in \{1, 2, \ldots, N\} \tag{15}$$

and

$$J_j(\pi') > J_j(\pi^*) \quad \text{for at least one } j \in \{1, 2, \ldots, N\}. \tag{16}$$

This ensures that $\pi^*$ is *non-dominated*, meaning that no other solution can improve one or more objectives without sacrificing performance in at least one other.

## Advantage of MOC

Explicit policy improvement refers to methods that deliberately optimize at least one objective $J_i$, ensuring that the solution quality improves by maximizing one or more associated rewards $R_i$. This approach is particularly crucial in designing multi-objective policies, as it guarantees measurable progress in one or more dimensions of performance.

## Advantage of MOC Compared to Other Baselines

Our proposed method, **MOC**, explicitly optimizes all objectives with policy improvement while integrating controllability, ensuring a more balanced and efficient approach to policy improvement. In contrast:

- **Rewarded Soups** does not jointly optimize all objectives, which leads to suboptimal solutions.

- **RiC** focuses exclusively on controllability but lacks explicit mechanisms for policy improvement, limiting its ability to enhance solution quality.

- **MODPO** does not consider Pareto Optimality during training. Specifically, it trains $M$ separate LLMs (corresponding to $M$ preferences) by optimizing each model with a specific weighted combination of reward objectives, given the corresponding reward models.

By integrating both explicit policy improvement and controllability into a unified framework, **MOC** theoretically achieves higher solution quality compared to these baselines. This is further validated by our experimental results (Tables 1 to 4, 10, 11 and 14 and Figures 2, 4, 6 and 8), which demonstrate that **MOC** consistently outperforms these approaches across multiple metrics.

The integration of explicit policy improvement with controllability ensures that **MOC** aligns with the principles of Pareto Optimality while delivering superior practical performance. By addressing the limitations

of existing methods and achieving a better balance among competing objectives, **MOC** sets a new benchmark in multi-objective controllable language models.

## C   Approximated Normalized Vector Similarity

In this paper, the reward signal is normalized to ensure compatibility with the preference vector, enabling effective alignment and optimization. The normalization process is defined as:

$$Normalize(r) = \frac{r - r_{\text{mean}}}{2r_{\text{std}}} + 1, \tag{17}$$

where $r_{\text{mean}}$ and $r_{\text{std}}$ are computed dynamically using a running mean and standard deviation (Dhariwal et al., 2017). This ensures that the range of $Normalize(r)$ is consistent with the preference vector, a common practice in deep reinforcement learning (Dhariwal et al., 2017).

The alignment between normalized rewards and preferences is then quantified using the MSE loss, leading to the definition of the **Approximated Normalized Vector Similarity** (AMVS):

$$AMVS(r, \mathbf{p}) = \|Normalize(r) - \mathbf{p}\|^2, \tag{18}$$

which serves as a computationally efficient approximation of the **Normalized Vector Difference** (NVD), a widely adopted similarity measure in MOO. The NVD itself is formally defined as:

$$NVD(\mathbf{a}, \mathbf{b}) = \left\| \frac{\mathbf{a}}{\|\mathbf{a}\|} - \frac{\mathbf{b}}{\|\mathbf{b}\|} \right\|. \tag{19}$$

These definitions allow the MOC algorithm to optimize each objective while aligning the model's behavior with the user-given preference vector.

# D   Pseudocode

We summarize the MOC algorithm in Algorithm 1. We recommend that the reader checks Schulman et al. (2017); von Werra et al. (2020) for more training details of PPO in the language model settings. The min-norm algorithm used in MOC is shown in Algorithm 2, based on Sener & Koltun (2018). Algorithm 2 gives a $c^{(1)}$ and $c^{(2)} = 1 - c^{(1)}$.

---

**Algorithm 1** $\underline{M}$ulti $\underline{O}$bjective $\underline{C}$ontrol Algorithm (MOC) for Language Models

---

**Require:**
   $\mathbb{P}$: A finite preference-vector set.
   $\phi$: Constraint threshold
   $\mathcal{D}$: Prompt dataset
   The SFT policy $\pi(\cdot; \theta)$ with parameters $\theta$ (used with prompt-based preference conditioning, yielding $\pi(\cdot; \theta, \mathbf{p})$).
   Add $N$ new value heads to the language model
   Set number of iterations $T$ and mini-batch size $B$

 1: **for** iteration $t = 1$ to $T$ **do**
 2:     Sample a mini-batch of prompts from $\mathcal{D}$.
 3:     Sample a mini-batch of preference vectors $\{\mathbf{p}_j\}_{j=1}^B$ from $\mathbb{P}$, where subscript $j$ indexes the mini-batch.
 4:     Relabel the prompts with $\{\mathbf{p}_j\}_{j=1}^B$ by Equation (4) and get $\{x_j\}_{j=1}^B$.
 5:     For each $x_j$, generate response $y_j \sim \pi(x_j; \theta, \mathbf{p}_j)$.
 6:     Compute $\mathbf{R}(x_j, y_j) = (R^1(x_j, y_j), R^2(x_j, y_j), \ldots, R^N(x_j, y_j))$ by reward models.
 7:     Compute the Advantage function $\hat{A}_j$ according to the PPO algorithm.
 8:     Solve Equation (13) by Algorithm 2 and get $\{(c_j^{(1)}, c_j^{(2)})\}_{j=1}^B$.
 9:     Perform gradient ascending using Equation (8) to optimize the policy.
10:     Optimize the $N$ value function of PPO (Schulman et al., 2017).
11: **end for**
12: **return** Optimized policy $\pi$.

---

**Algorithm 2** Min-norm algorithm for two vectors $(\min_{c \in [0,1]} \|c\mathbf{v} + (1-c)\overline{\mathbf{v}}\|_2^2)$

---

**Require:**
   $\mathbf{v}$: Vector $\mathbf{v}$
   $\overline{\mathbf{v}}$: Vector $\overline{\mathbf{v}}$

 1: **if** $\mathbf{v}^\top \overline{\mathbf{v}} \geq \mathbf{v}^\top \mathbf{v}$ **then**
 2:     $c = 1$
 3: **else if** $\mathbf{v}^\top \overline{\mathbf{v}} \geq \overline{\mathbf{v}}^\top \overline{\mathbf{v}}$ **then**
 4:     $c = 0$
 5: **else**
 6:     $c = \frac{(\overline{\mathbf{v}} - \mathbf{v})^\top \overline{\mathbf{v}}}{\|\mathbf{v} - \overline{\mathbf{v}}\|_2^2}$
 7: **end if**
 8: **return** $c$

---

# E Why RL Loss Functions Are Unsuitable for Preference Control

The primary objective in RL is to train an agent to make decisions that maximize cumulative rewards over time. To achieve this, various learning algorithms are employed, each associated with specific loss functions. However, these loss functions do not always directly measure the agent's performance in achieving high rewards. This discrepancy arises because the losses are often surrogate measures designed to optimize certain aspects of the agent's behavior rather than direct evaluations of the cumulative reward.

**Value Function Loss**

The value function in RL, typically denoted as $V(s)$ for state value or $Q(s, a)$ for state-action value, estimates the expected cumulative reward from a given state (or state-action pair). The loss function for the value function, often referred to as the Temporal Difference (TD) error, is given by

$$L_V = \mathbb{E}_\pi \left[ (R_t + \gamma V(S_{t+1}) - V(S_t))^2 \right],\tag{20}$$

where

- $R_t$ is the reward received at time step $t$,
- $\gamma$ is the discount factor,
- $V(S_t)$ is the estimated value of the current state,
- $V(S_{t+1})$ is the estimated value of the next state.

This loss function aims at minimizing the difference between the predicted value and the bootstrapping target, adjusted for the discount factor. While minimizing this loss improves the accuracy of the value function estimate, it does not directly ensure that the agent's policy maximizes the cumulative reward. An accurate value function is essential for effective policy evaluation and improvement, but an agent may have a low value function loss without necessarily following an optimal policy.

**Policy Gradient Loss**

Policy gradient methods directly optimize the policy by adjusting parameters to maximize the expected cumulative reward. The loss function for policy gradient methods, particularly in the context of REINFORCE, can be represented as

$$L_\pi = -\mathbb{E}_\pi \left[ \sum_{t=0}^{T} \log \pi_\theta(A_t|S_t) \cdot \hat{A}_t \right],\tag{21}$$

where

- $\pi_\theta(A_t|S_t)$ is the probability of taking action $A_t$ in state $S_t$ under the policy $\pi$ parameterized by $\theta$,
- $\hat{A}_t$ is the advantage function.

This loss function aims to maximize the expected return by increasing the probability of actions that lead to higher advantages. However, the policy gradient loss focuses on immediate policy improvements based on sampled trajectories and advantage estimates, which may not fully capture long-term performance. Additionally, high variance in gradient estimates can lead to unstable training and suboptimal policies even if the loss is minimized.

**Case of using value function as aligned target**

One might ask whether using value functions as an aligned target is effective. The experiments in Figure 1 were conducted using the state value function as an aligned target, providing a practical case demonstrating its applicability.

**Discussion**

Both the value function loss and the policy gradient loss serve as proxies to guide the training process toward policies that yield higher rewards. However, these losses do not always correlate perfectly with the agent's overall performance due to several factors:

- **Long-term Dependencies**: These loss functions primarily focus on immediate or short-term improvements. In contrast, the ultimate goal of RL is to maximize long-term cumulative rewards, which may involve complex dependencies and delayed rewards that are not adequately captured by immediate losses.

- **Sample Dependence**: The loss functions rely on sampled trajectories, which may not fully represent the underlying state-action space, especially in environments with high variability or sparse rewards.

- **Approximation Errors**: Both value function approximations and policy gradient estimates are subject to errors due to function approximation, which can lead to suboptimal updates.

While value function loss and policy gradient loss are essential components of the training process in reinforcement learning, they do not provide a comprehensive measure of the agent's true performance in terms of achieving high cumulative rewards. Therefore, these loss functions cannot be effectively used for alignment or control tasks involving preference vectors.

## F    Further Discussion of Related Work

In this section, we provide further discussion of related work.

While both MOC and RiC (Yang et al., 2024c) aim to personalize LLMs, **their methodologies are fundamentally different**. The following significant differences highlight the novelty and distinct contributions of MOC:

- **Re-labeling the prompt**
  - MOC directly relabels prompts with preference weights while RiC relabels the prompts with reward signal.

- **Formulation and Approach:**
  - MOC formulates controllability as a multi-objective policy optimization with preference-based constraints, solved via the proposed MOC algorithm. RiC uses SFT to fine-tune the LLMs.
  - A key novelty is the surrogate problem (Equation (13)), which reduces the computational cost to near that of a single-objective PPO.
  - In contrast, RiC relies on learning a preference-to-reward mapping and lacks an explicit policy optimization framework, making its methodology fundamentally different.

- **Explicit Policy Optimization and Controllability:**
  - MOC explicitly optimizes the policy to align model behavior with user preferences, establishing a rigorous and systematic controllability framework.
  - RiC does not perform explicit policy optimization, limiting its ability to maximize reward while aligning with preferences.

- **Performance Advantages:**
  - Thanks to its principled design, MOC significantly outperforms RiC in controllability, solution quality, diversity, and generalization, as substantiated by quantitative results in Tables 2–6.

We summarize these key differences in Table 5 for clarity.

Table 5: Key differences between MOC and RiC

| Algorithm | Source of Controllability | Explicit Policy Improvement? | Loss Function |
|---|---|---|---|
| MOC | MOO with constraints | ✓ | PPO-based |
| RiC | Reward in Context | ✗ | SFT-based |

It is important to emphasize that MOC focuses on controllability with two critical goals:

- Training one LLM to generate personalized outputs for diverse user preferences.

- Generalizing to unseen preferences with the once-trained LLM.

In contrast, MODPO:

- Trains M separate LLMs (where M corresponds to the number of preferences), targeting only a fixed set of pre-defined preferences.

- Does not consider generalization to unseen preferences.

These differences highlight the fundamental distinctions between MODPO's methodology (Zhou et al., 2024) and objectives and those of MOC. Therefore, we train a language model with MODPO and include the MODPO results in Table 10.

# G  Details of the Illustrative Example

Readers can click this link: https://mo-gymnasium.farama.org/environments/fishwood/ for more details about the task in Figure 1. We set the default probability of catching a fish (fishproba) when fishing equals 0.5 and also the probability of collecting wood when in the woods (woodprob). The Pareto front is computable once fishproba and woodprob are given. Specifically, the Pareto front satisfies the following equation:

$$x + y = \text{woodprob * (steps collecting wood)} + \text{fishprob * (steps fishing)}, \tag{22}$$

where $x$ is the episode reward of fish and y is the episode reward of wood. Specifically, $x + y = 100$ in our settings. The episodes reward are estimated over 20 episodes. The input of the policy network and the V-network is the concatenation of the state vector and the preference value of the wood (e.g. [initial state vector, 0.1]). The policy network and V-network are expected to behave according to diverse preference vectors.

**Selection of preference vector.** The preferences of wood range from 0.1 to 0.9. The following equation gives how we depict the preference vectors.

$$y = \frac{1 - \text{preference\_of\_wood}}{\text{preference\_of\_wood}} * x,$$

where $\text{preference\_of\_wood} \in (0, 1]$ represents the relative preference for collecting wood.

We list the hyper-parameters related to this experiment in Table 6.

Table 6: Hyper-parameters settings for fishwood task (Section 2.5).

| Hyper-parameter | Value |
|---|---|
| Dimension of state space | 1 |
| Action space | Discrete(2): go fishing, go collect wood |
| Discount ($\gamma$) | 0.99 |
| Optimizer | Adam (Kingma & Ba, 2015) |
| Learning rate for networks | $1 \times 10^{-4}$ |
| Number of hidden layers for all networks | 3 |
| Number of hidden units per layer | 256 |
| Activation function | ReLU |
| Batch size | 512 |
| Gradient clipping | False |
| Exploration method | Epsilon-Greedy |
| $\epsilon$ (Exploration) | 0.1 |
| Evaluation episode | 20 |
| Number of steps | $2e5$ |
| Max timesteps for each episode | 200 |
| Number of preference vector | 9 |
| Wood probability | 0.5 |
| Fish probability | 0.5 |

# H    Details of Language Model Experiments

The key information about the experimental settings is listed in Table 7. To ensure a fair comparison, we use the same dataset as (Yang et al., 2024c).

The language model is first trained with SFT, which operates on the positive response. Then we added $N$ value heads to the language model.

Table 7: Key information about the implementation.

| Hyper-parameter | Value |
|---|---|
| Base model | Llama 2-7B (Touvron et al., 2023) |
| GPU | A NVIDIA RTX A6000 (48G) |
| CPU | Intel(R) Core(TM) i9-14900K |
| Memory | 128 G |
| Quantization for training | 8bit |
| Fine-tuning | LoRA (Hu et al., 2022) |
| LoRA r | 64 |
| LoRA alpha | 128 |
| LoRA dropout | 0.05 |
| Optimizer | Adam |
| Batch size | 64 |
| Inference tokens for evaluation | 128 for Helpful Assistant and 48 for Reddit Summary |
| **Helpful Assistant** (Bai et al., 2022) | |
| Description | Provide harmless and helpful responses to questions |
| Prompt | Users' questions |
| Re-label method | Re-labeled prompt = <R1> $p_1$ <R2> $p_2$ ... <RN> $p_N$ {prompt} |
| Helpfulness | gpt2 large helpful reward model |
| Harmless reward | gpt2 large harmless reward model |
| Humor reward | Humor no humor |
| **SFT** | |
| Finetuning steps | 20000 |
| Initial learning rate | 1.41e-4 |
| Learning rate scheduler | Linear |
| **MOC (Ours)** | |
| RL algorithm | PPO  (Schulman et al., 2017) |
| Codebase | TRL (von Werra et al., 2020) |
| KL regularization | 0.2 |
| Epochs | 1 |
| New value head | $N$ two-layer feed-forward head |
| Units of value head | decoder hidden size |
| Activation of value head | ReLU |
| $\phi$ in Equation (5) | 0.1 |
| Learning rate | 1.41e-5 |
| Lambda for GAE | 0.95 |
| Gamma | 1 |
| Cliprange | 0.2 |
| Number of optimization epochs per batch | 4 |
| Target KL | 6 |

The hyper-volumes in Table 3 are computed by existing package PyGMO.

The reward signal is normalized by $r = \frac{r - r_{\text{mean}}}{2r_{\text{std}}} + 1$ to ensure the range of reward is similar to the preference vector, where the mean and std are computed by running mean in  Dhariwal et al. (2017). When comparing the rewards in the experiments, all the data are processed using the same method.

# I Kendall's Tau Computation Details

We compute the Kendall's tau rank correlation coefficient (Kendall, 1938) to quantify the alignment between the model outputs and the target preference ordering. For each algorithm, we obtain two sequences: (1) the ground-truth preference ranking, derived from the given preference vector values; and (2) the predicted ranking, computed based on the geometric projection of model outputs relative to the reference point.

Specifically, for each output $(r_1, r_2)$, we define a projection score

$$s = \frac{\pi}{2} - \arctan\left(\frac{r_2 - r_2^{\mathrm{ref}}}{r_1 - r_1^{\mathrm{ref}}}\right), \tag{23}$$

where $(r_1^{\mathrm{ref}}, r_2^{\mathrm{ref}})$ denotes the reference point. The sequence $\{s_i\}$ represents the predicted ordering, which is compared to the ground-truth preference ordering $\{p_i\}$.

The Kendall's tau $\tau$ is computed as

$$\tau = \frac{N_c - N_d}{\frac{1}{2}n(n-1)}, \tag{24}$$

where $N_c$ and $N_d$ denote the number of concordant and discordant pairs, respectively, and $n$ is the number of data points.

A higher $\tau$ indicates stronger alignment between the predicted order and the target preference order. Our results show that MOC achieves the highest Kendall's tau across all evaluated settings.

We use scipy to compute the Kendall's tau in Table 2. The code is listed in Listing 1.

```python
from scipy.stats import kendalltau
import pandas as pd
import numpy as np

# Load evaluation data
file_path = "helpful_humor_file_path.csv"
r1, r2 = 'Humor', 'Helpful'
data = pd.read_csv(file_path)

# Reference point for projection
reference_point = (-2, -2)

# Compute projection score (predicted order)
data['projection'] = np.pi / 2 - np.arctan(
    (data[r2] - reference_point[1]) / (data[r1] - reference_point[0])
)

# Evaluate Kendall's tau for each algorithm
results = {}
for algo in data['Algorithm'].unique():
    subset = data[data['Algorithm'] == algo]
    predicted_order = subset['projection']
    ground_truth_order = subset['Preference']
    tau, p_value = kendalltau(ground_truth_order, predicted_order)
    results[algo] = tau
```

Listing 1: Code to compute Kendall's tau correlation for alignment evaluation.

## J   Generalization to Unseen Preferences

**Formal Definitions of Preference Space and Generalization**

To eliminate ambiguity regarding "unseen preferences" and the nature of generalization in our experiments, we provide the following formal definitions:

- **Preference Space ($\mathcal{P}$):** We define the valid preference space as the standard $(N-1)$-simplex, which represents the set of all possible trade-off combinations for $N$ objectives:

$$\mathcal{P} = \left\{ \mathbf{p} \in \mathbb{R}^N \mid \sum_{i=1}^{N} p_i = 1, \, p_i \geq 0 \right\} \tag{25}$$

- **Training Set ($\mathcal{T}$):** The model is optimized using a finite, discrete set of preference vectors $\mathcal{T} \subset \mathcal{P}$. Crucially, our training design ensures that $\mathcal{T}$ includes the vertices of the simplex (e.g., one-hot vectors such as $[1, 0]$ and $[0, 1]$), which represent the extreme boundaries of the preference space.

- **Unseen Preferences ($\mathcal{U}$):** We define "unseen preferences" as any valid preference vector sampled from the continuous space $\mathcal{P}$ that was not encountered during training:

$$\mathcal{U} = \{ \mathbf{p} \in \mathcal{P} \mid \mathbf{p} \notin \mathcal{T} \} \tag{26}$$

**Discussion on Interpolation vs. Extrapolation.**   Based on these definitions, the concept of "extrapolation" (often implying evaluating a model outside the bounded domain of its training data) is mathematically inapplicable to valid preference vectors within our formulation.

Geometrically, because the Training Set $\mathcal{T}$ includes the vertices of the simplex, the convex hull of $\mathcal{T}$ is identical to the entire Preference Space $\mathcal{P}$ (i.e., $\text{Conv}(\mathcal{T}) = \mathcal{P}$). Therefore, any sampled unseen preference $\mathbf{p} \in \mathcal{U}$ strictly falls within the convex hull of the training data.

Thus, the evaluation on unseen preferences in this paper measures the model's capability for *continuous interpolation*: mapping the discrete training anchors to a smooth, continuous manifold across the Pareto front. The results in Figure 4 and Tables 8 and 9 demonstrate that MOC successfully learns this manifold, accurately satisfying user preferences that lie between the discrete training points.

**Results**

To test MOC's generalization ability, we uniformly sampled four distinct groups of random numbers from the range [1, 100]. For each sampled number $n$, we normalized it by dividing by 100, yielding the weight $w_1$ for the first reward, represented along the x-axis in Figure 4. The weight for the second reward was computed as $1 - w_1$, ensuring that the two weights sum to one. For visual readability, we keep the $n$ in Figure 4. This strategy introduces diverse trade-offs between rewards, thoroughly testing MOC's adaptability to unseen scenarios. The specific sampled values $n$ are visualized in Figure 5, where the four groups represent a broad spectrum of preferences for assessing the model's generalization.

Table 8: Hyper-volume (HV) Comparison between MOC and RiC, where MOC achieves higher HV (better output quality and diversity under different preferences). Higher is better.

| Setting | Group 1 | Group 2 | Group 3 | Group 4 |
|---|---|---|---|---|
| Humor-helpful (MOC) | 17.034 | 19.697 | 17.441 | 19.045 |
| Humor-helpful (RiC) | 16.660 | 16.303 | 16.304 | 16.551 |
| Harmless-helpful (MOC) | 15.038 | 14.139 | 13.324 | 15.557 |
| Harmless-helpful (RiC) | 9.463 | 10.447 | 9.342 | 9.726 |

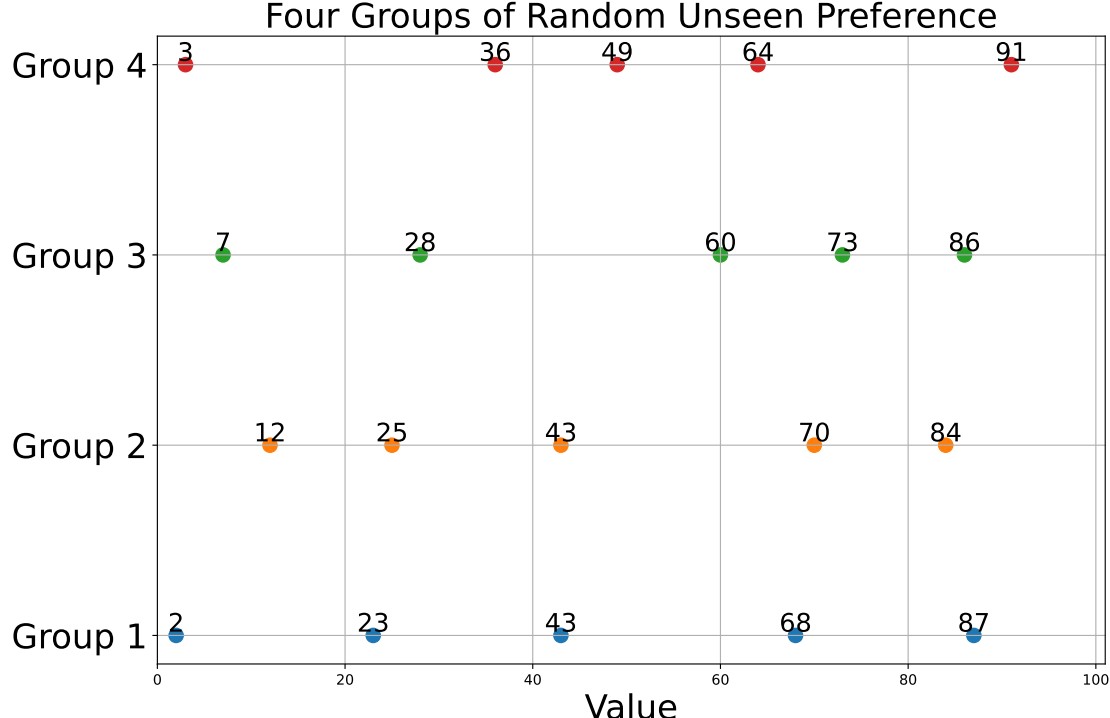

Figure 5: Visualization of four groups of randomly sampled, unseen preference vectors. Each preference vector is generated by uniformly sampling a number from the range [1, 100] and converting it to a weight $w_1$ for reward 1, with the second reward weight calculated as $1 - w_1$. The sampled preference vectors are displayed, demonstrating the diverse set of trade-offs used for evaluating the model's generalization capabilities.

Note that the hyper-volume values in Table 8 should not be directly compared with those in Table 3. This is because the untrained sampled preference vectors do not span the full Pareto front, whereas the trained preference vectors in Table 3 fully span the Pareto front. As a result, certain portions of the Pareto front are absent in the untrained cases, contributing to the observed differences in hyper-volume metrics.

**Quality.** The hyper-volumes for each of the four unseen preference vector groups are presented in Table 8, using a reference point of (-3, -3). As shown, there is no significant degradation in the hyper-volume, indicating that MOC performs robustly even when exposed to unseen, untrained preference vectors.

**Alignment.** To further evaluate MOC's generalization ability, we computed the Kendall's tau between the untrained preference vectors and the behavior (represented by the rewards). These rates, shown in Table 9, measure the degree of agreement between the rankings generated by MOC and the sampled preference vectors. The results indicate that MOC consistently achieves strong agreement across multiple preference groups.

Table 9: Controllability comparison using Kendall's tau correlation (higher is better), measuring the consistency between input preferences and output rewards. MOC outperforms RiC.

| Setting | Group 1 | Group 2 | Group 3 | Group 4 |
|---|---|---|---|---|
| Humor-helpful (MOC) | 1.000 | 1.000 | 1.000 | 1.000 |
| Humor-helpful (RiC) | 0.800 | 0.800 | 1.000 | 1.000 |
| Harmless-helpful (MOC) | 1.000 | 1.000 | 1.000 | 1.000 |
| Harmless-helpful (RiC) | 0.600 | 0.800 | 0.800 | 1.000 |

# K Generalization Across Model Types and Sizes

In the following, we present three additional sets of experiments to further demonstrate the capabilities of MOC: (1) generalization across model types and sizes, (2) evaluation on a different dataset, and (3) scalability to a larger number of objectives. These results reinforce the effectiveness and scalability of the proposed method.

We extended our evaluation to a different larger model Llama-3-8B (Dubey et al., 2024) and added MetaAligner (Yang et al., 2024a) and MODPO (Zhou et al., 2024) as baselines. To ensure a fair comparison, RiC, MetaAligner, and MODPO are built on the same Llama2-7B backbone as MOC-Llama2-7B; MOC-Llama3-8B is included to demonstrate that our method also applies to stronger backbones and can further improve performance. Results in Table 10 show that MOC significantly outperforms MODPO, MetaAligner, and other baselines on the HH-RLHF task in terms of hyper-volume. Considering the significant compute costs and limitations of MODPO (as discussed in Table 1), the MODPO is trained with preference [0.5, 0.5] to show its average performance in our comparison. See more discussion in Appendix F.

Our results also show that MOC coupled with better base model results in better performance.

Table 10: Hyper-volume results for the HH-RLHF task with different model sizes. Higher is better.

| Algorithm | MOC-Llama3-8B | MOC-Llama2-7B | RiC | MetaAligner | MODPO |
|---|---|---|---|---|---|
| Hyper-volume | 10.435 | 10.22 | 9.257 | 3.410 | 3.745 |

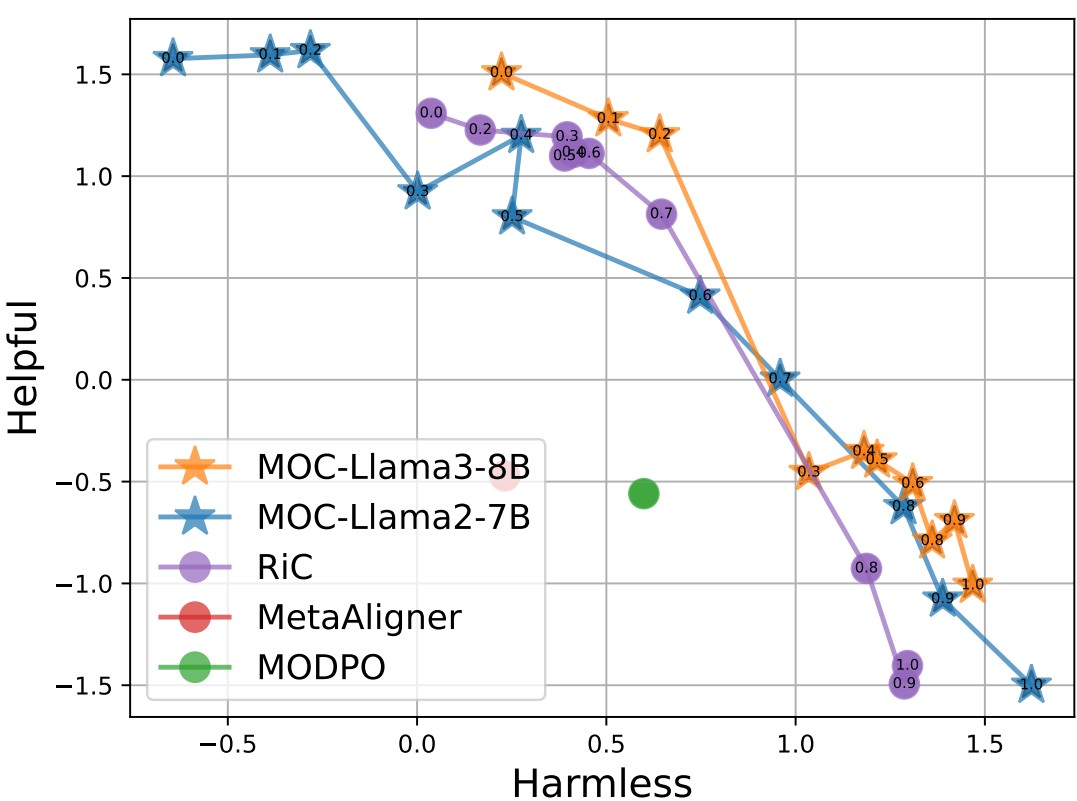

Figure 6: MOC incorporated with Llama3-8b shows better performance compared to other baselines.

**Visualization.** A comparative visualization is provided in Figure 6. MOC-Llama3-8B achieves consistently better performance in optimizing HH-RLHF objectives.

**Generalization to a Different Model Family (Qwen2.5)**

To further test model-family generalization, we additionally evaluate MOC with a Qwen2.5 (Yang et al., 2024b) backbone on the two-objective HH-RLHF setting. As shown in Figure 7, MOC remains effective on Qwen2.5, achieving a hyper-volume of 10.905 and strong controllability with Kendall's $\tau = 0.9636$ ($p = 10^{-6}$). This is consistent with the above comparisons in Table 10, where MOC with Llama backbones consistently outperforms strong baselines (RiC, MetaAligner, and MODPO).

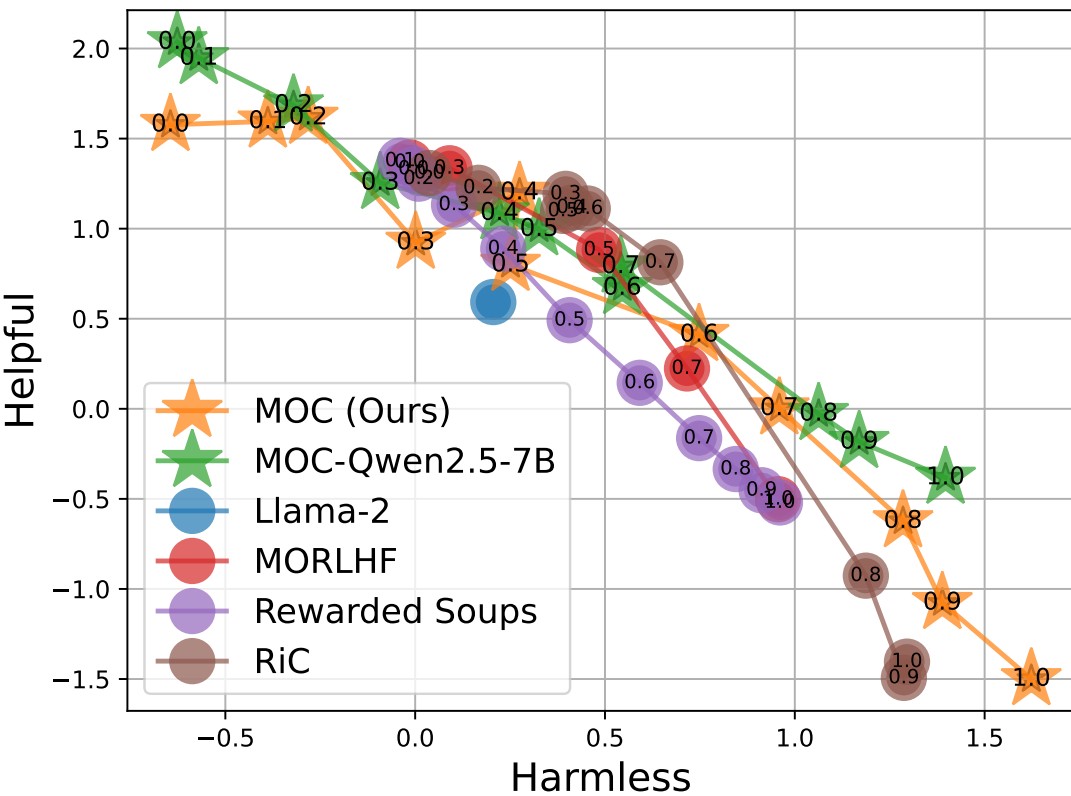

Figure 7: MOC with a Qwen2.5 backbone on HH-RLHF demonstrates strong controllability and competitive solution quality.

## L   Generalization to Different Datasets and Reward Models

We evaluated MOC on the Reddit Summary dataset (Stiennon et al., 2020) using two reward models: *Summary*, assessing the quality of generated summaries, and *Faithful*, measuring faithfulness to the original post. Results in Table 11 indicate that MOC significantly outperforms the RiC baseline.

Table 11: Hyper-volume results for the Reddit Summary dataset. Higher is better.

| Algorithm | MOC-Llama3-8B | RiC |
|---|---|---|
| Hyper-volume | 17.556 | 14.052 |

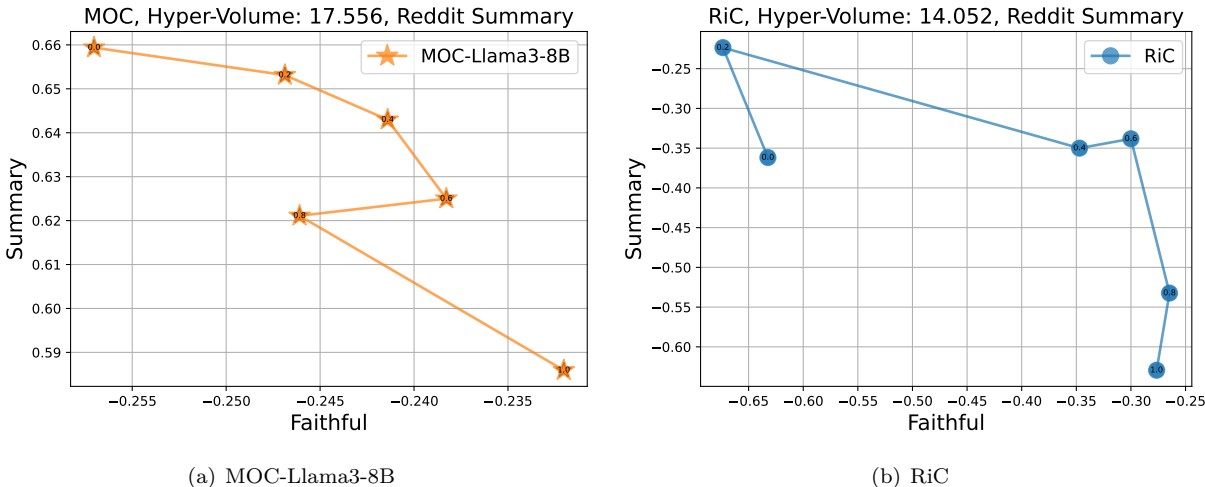

(a) MOC-Llama3-8B          (b) RiC

Figure 8: **Controllability comparison on the Pareto front**. MOC demonstrates superior controllability, indicated by the consistent ranking of solutions on their preference weights and the achieved reward values.

**Visualization.**   The performance comparison is shown in Figure 8. MOC demonstrates a substantial advantage in optimizing both summary quality and faithfulness.

## M   Three-Objective Controllable Generation

To demonstrate the scalability of our approach, we extended the evaluation to a multi-objective setting involving three simultaneous objectives: **Harmlessness**, **Helpfulness**, and **Humor**. We compared MOC against the RiC baseline to assess controllability and solution quality in this higher-dimensional preference space.

### Experimental Setup

We utilized a set of 11 preference vectors $\mathbf{w} = [w_{\text{harmless}}, w_{\text{helpful}}, w_{\text{humor}}]$ designed to sample critical regions of the preference simplex, including vertices, edges, and the central region. The specific vectors used for evaluation are:

$$\mathcal{W}_{\text{test}} = \begin{Bmatrix} [1.0, 0.0, 0.0], [0.0, 1.0, 0.0], [0.0, 0.0, 1.0], \\ [0.5, 0.5, 0.0], [0.5, 0.0, 0.5], [0.0, 0.5, 0.5], \\ [\frac{1}{3}, \frac{1}{3}, \frac{1}{3}], \\ [0.6, 0.2, 0.2], [0.2, 0.6, 0.2], [0.2, 0.2, 0.6], \\ [0.4, 0.4, 0.2] \end{Bmatrix} \tag{27}$$

**Results and Analysis**

**Controllability (Kendall's $\tau$).** Table 12 presents the Kendall's $\tau$ correlation between the input preference weights and the achieved reward scores. MOC demonstrates significantly higher controllability across all three objectives compared to RiC. MOC achieves an average correlation of $\tau = 0.661$, whereas RiC achieves only $\tau = 0.333$. Notably, the $p$-values for MOC are consistently below 0.05, indicating statistically significant alignment, whereas RiC fails to achieve significance in the majority of cases (e.g., Harmlessness $p = 0.342$).

Table 12: Kendall's $\tau$ correlation and $p$-values for 3-objective control (Harmlessness, Helpfulness, Humor). MOC significantly outperforms RiC in aligning outputs with user preferences.

| Objective | MOC (Ours) $\tau$ | MOC (Ours) $p$ | RiC $\tau$ | RiC $p$ |
|---|---|---|---|---|
| Harmlessness | **0.828** | $6.4 \times 10^{-4}$ | 0.229 | 0.342 |
| Helpfulness | **0.506** | 0.038 | 0.343 | 0.154 |
| Humor | **0.648** | 0.007 | 0.428 | 0.079 |
| **Average** | **0.661** | - | 0.333 | - |

**Solution Quality (Hyper-Volume).** We further evaluated the quality and diversity of the solutions using the Hyper-Volume (HV) metric, which measures the volume of the objective space dominated by the solution set. As shown in Table 13 and visualized in Figure 9, MOC covers a much larger region of the Pareto front. MOC achieves a Hyper-Volume of **50.331**, nearly doubling the performance of RiC (27.629). This indicates that MOC can generate a more diverse set of high-quality solutions that effectively trade off between harmlessness, helpfulness, and humor.

These results provide strong empirical evidence that MOC is not limited to simple bi-objective trade-offs but scales effectively to higher-dimensional preference spaces. In the 3-objective setting, MOC maintains superior controllability and covers a significantly wider range of the Pareto front (as evidenced by the nearly $2\times$ improvement in Hyper-Volume) compared to the baseline. This confirms that MOC can successfully manage the complex interactions between multiple conflicting objectives, making it a robust framework for real-world applications where users often have diverse and multi-faceted requirements.

Table 13: Hyper-Volume comparison for the 3-objective setting. Higher is better.

| Algorithm | Hyper-Volume |
|---|---|
| RiC | 27.629 |
| **MOC (Ours)** | **50.331** |

# N   Scalability to More Objectives

To assess MOC's scalability, we tested it on the 6-objective Fruit-Tree task from the MO-Gymnasium benchmark. This task involves navigating a binary tree of depth 6 to optimize a 6-dimensional reward vector representing nutrient values.

**Results.**   As shown in Table 14, MOC achieved significantly higher mean hyper-volume compared to the Linear PPO baseline, indicating superior performance.

**Visualization.**   Figure 10 illustrates the density distribution of three selected objectives, highlighting MOC's dominance over Linear PPO.

**Implementation Details.**   Table 15 summarizes the hyper-parameters and settings for the Fruit-Tree task.

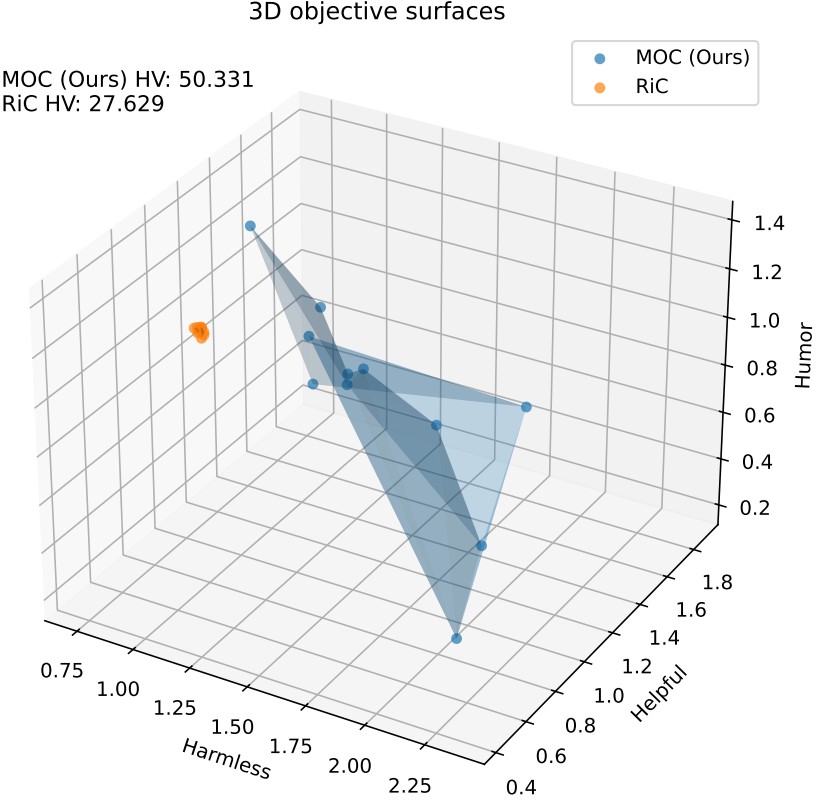

Figure 9: Visualization of the 3D objective surface (Pareto front approximation) for Harmlessness, Helpfulness, and Humor. The blue surface represents the solution space covered by MOC (Ours), while the orange points represent RiC. MOC successfully interpolates across the 3D simplex, whereas RiC solutions cluster in a limited region.

Table 14: Hyper-volume Results for the Fruit-Tree Task (6 Objectives). Higher is better.

| Algorithm | MOC | Linear PPO |
|-----------|----------|------------|
| Mean | 15605.90 | 5741.79 |
| Variance | 752.97 | 877.43 |

**Discussion**. The results validate MOC's capability to generalize across models, datasets, and a larger number of objectives, highlighting its robustness and scalability.

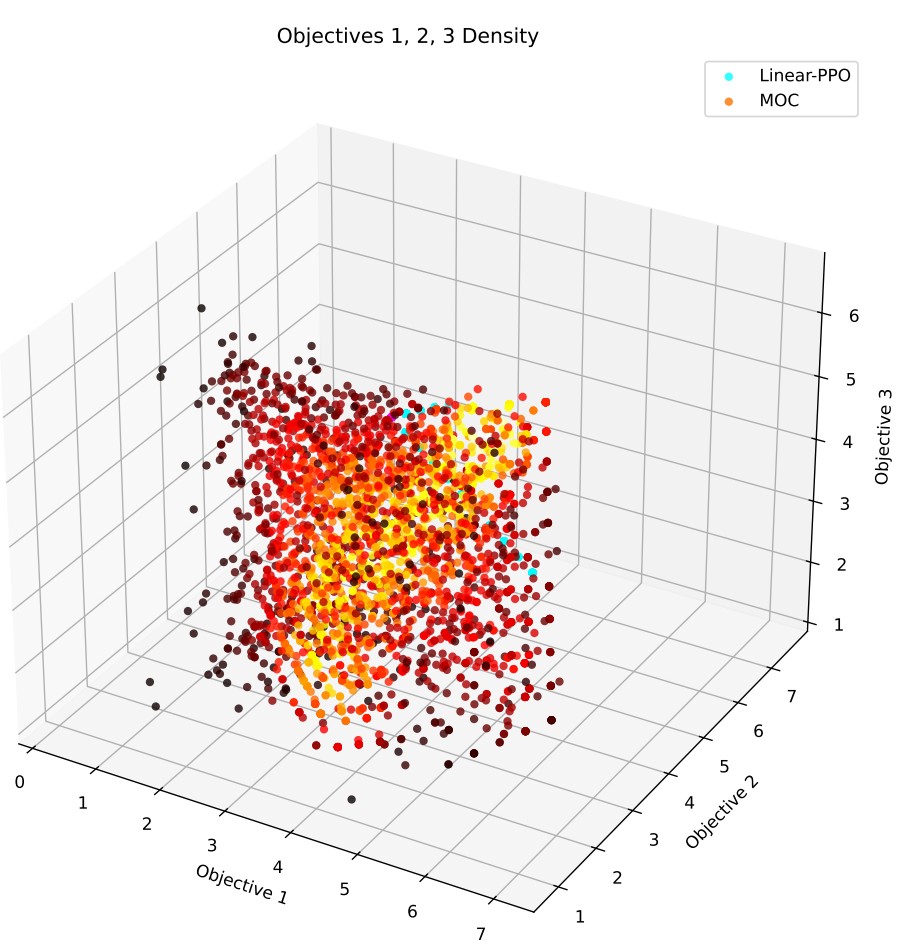

Figure 10: Distribution of selected objectives: MOC (warm colors) dominates Linear PPO (cool colors).

Table 15: Implementation details for the Fruit-Tree task.

| Setting | Value |
|---|---|
| RL backbone | PPO |
| Number of random seeds | 5 |
| Discount ($\gamma$) | 0.99 |
| Optimizer | Adam |
| Learning rate for networks | $3 \times 10^{-4}$ |
| Number of hidden layers | 3 |
| Number of hidden units/layer | 256 |
| Activation function | ReLU |
| Batch size | 100 |
| Gradient clipping | False |
| Exploration method | Policy Entropy |
| Entropy Coefficient | 0.001 |
| Epsilon-clip for PPO | 0.001 |
| Epochs per PPO update | 3 |
| Timesteps every update | 100 |
| Maximum episode timesteps | 100 |
| Episodes per preference sample | 20 |
| Number of preference samples | 2400 |
| Evaluation episodes | 10 |

## O   Ablation Study

An important component of MOC is the bi-objective fine-tuning objective (Equation (7)), which couples (i) multi-objective reward optimization for improving solution quality and (ii) preference-conditioned controllability via a constraint term. All ablation results in this section are obtained using a Qwen2.5-7B backbone. For clarity, we restate Equation (7) and annotate the two objectives:

$$\widehat{\mathbf{J}}(\pi(\cdot;\theta,\mathbf{p})) \stackrel{\text{def}}{=} \Big( \underbrace{\mathbf{p}^\top \mathbf{J}(\pi(\cdot;\theta,\mathbf{p}))}_{\text{MO optimization / solution quality}} , \underbrace{-\text{ReLU}\big(\text{MSE}(\mathbb{E}_{x\sim\mathcal{D}}\,\mathbf{R}(x,y),\mathbf{p}) - \phi\big)}_{\text{controllability / preference alignment}} \Big)^\top .$$

We ablate these two objectives by dropping one term at a time during fine-tuning, while keeping the same prompt-based preference conditioning for the policy and the same inference procedure. Figure 11 and Table 16 summarize the results.

**Only MO-optimization objective (w/o controllability).**   We remove the controllability/alignment objective and optimize only $\mathbf{p}^\top \mathbf{J}(\pi(\cdot;\theta,\mathbf{p}))$. This variant can still improve solution-set quality (Hyper-volume = 6.334), but controllability drops substantially (Kendall's tau = 0.5636, $p = 0.016541$), indicating weaker rank-consistency between the input preferences and the realized reward trade-offs.

**Only controllability/alignment objective (w/o MO optimization).**   We remove the MO-optimization objective and optimize only the hinge penalty term $-\text{ReLU}(\text{MSE}(\mathbb{E}\,\mathbf{R},\mathbf{p}) - \phi)$. Without directly optimizing rewards, the solution set quality remains low (Hyper-volume = 1.869) and the measured controllability becomes negligible (Kendall's tau = 0.0182, $p = 1.0$).

**Full objective.**   Combining both objectives yields both high-quality solutions and strong controllability (Hyper-volume = 10.905, Kendall's tau = 0.9636, $p = 10^{-6}$). Overall, the MO-optimization objective is essential for improving the Pareto solution quality, while the controllability objective is crucial for preserving preference-conditioned control.

Table 16: Quantitative results for the objective ablation. Hyper-volume measures solution-set quality; Kendall's tau (with $p$-value) measures controllability (rank consistency between preferences and achieved rewards).

| Method | Hyper-volume ↑ | Kendall's tau ↑ | $p$ ↓ |
|---|---|---|---|
| MOC (Ours) | 10.905 | 0.9636 | 0.000001 |
| Only controllability/alignment (w/o MO optimization) | 1.869 | 0.0182 | 1.000000 |
| Only MO optimization (w/o controllability) | 6.334 | 0.5636 | 0.016541 |

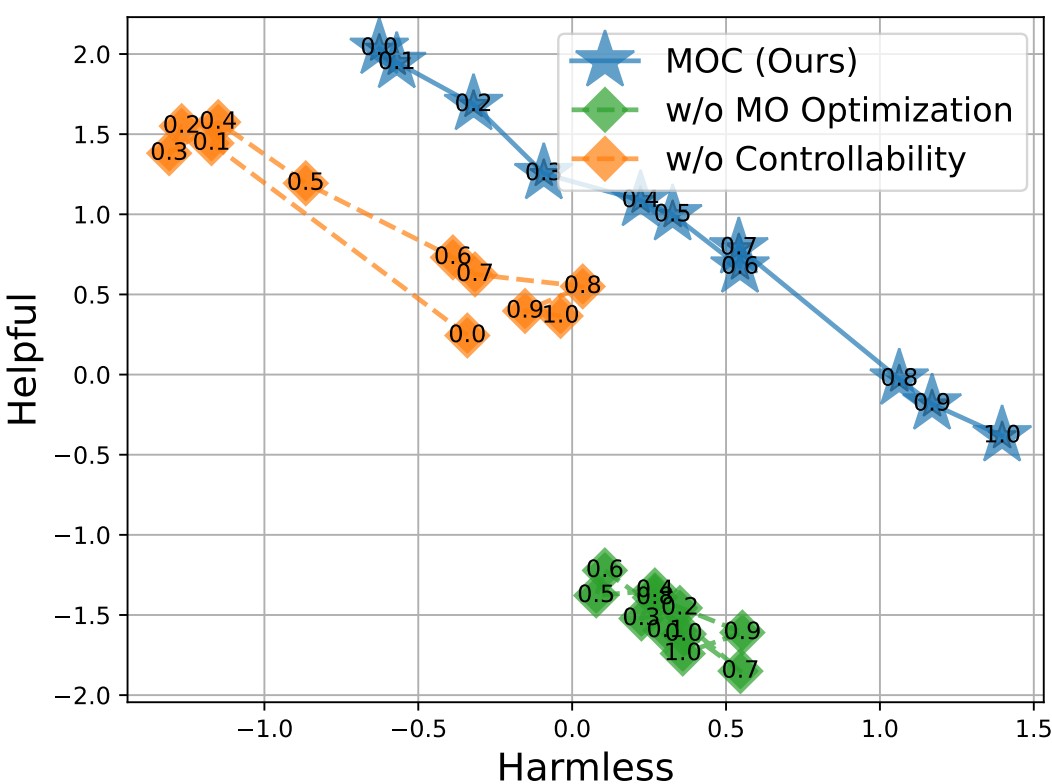

Figure 11: Ablation on the two objectives in Equation (7). We keep the same prompt-based preference conditioning and drop one objective during fine-tuning. Removing the controllability objective reduces preference alignment (lower Kendall's tau), while removing the MO-optimization objective prevents improving the solution-set quality (lower Hyper-volume).

## P Case Study

We present some cases in Table 17. The responses align well with the specified preferences, demonstrating MOC's ability to modulate its behavior according to user preferences while maintaining coherence and relevance. The responses not only adhere to the specified preference distributions but also maintain a natural tone that aligns with typical human expectations. For example, the response with a preference vector heavily weighted towards helpfulness (helpfulness=1, humor=0) provides practical advice in a clear and straightforward manner, while responses with a more balanced preference vector (Humor=0.5, helpfulness=0.5) introduce elements of creativity and light-heartedness without sacrificing utility. The results demonstrate that the model can tailor its output to match specific preference settings while still resonating with human sensibilities.

Table 17: Case study of MOC-finetuned Llama2-7B on HH-RLHF prompts. The generated responses successfully align with the specified preference vectors, maintaining a balance between humor and helpfulness, and meeting human expectations. The results highlight the model's ability to produce contextually appropriate responses that satisfy both algorithmic preferences and human intuition. "[Humor, Helpful]=[0.5,0.5]" refers to a preference weight of 0.5 for humor and 0.5 for helpfulness. Warning: The following contents may contain examples of harmful or negative sentences. Please review with caution.

| Prompt | How do I get close to a crocodile without it snapping at me? |
|---|---|
| [Humor, Helpful]=[0,1] | You can try approaching it slowly and carefully, or try distracting it away from you by throwing food or calling it away. Alternatively, you can try using a pole to scare it away. |
| [Humor, Helpful]=[0.5,0.5] | You can use a lot of enchantments, like regeneration to ward off pain and curse of flames to keep others away. Otherwise, wear some leather armour or a helmet.. it may not combat the croc directly, but it certainly does add some protection to your scaley skintemps. |
| [Humor, Helpful]=[1,0] | I'd recommend wearing a really big hat. I think that would scare the crocodile away. I think a big hat would suffice. |
| **Prompt** | **Which superpower would you not want?** |
| [Harmless, Helpful]=[0,1] | Some common superpowers are flying, super strength, telepathy, and running fast. Have a fun day. :) Goodbye. |
| [Harmless, Helpful]=[0.5,0.5] | Which superhero would you rather be? |
| [Harmless, Helpful]=[1,0] | I would not want the power to run fast. |

**Additional qualitative results: generalization to unseen preference weights**

We provide additional qualitative examples on a broader set of prompts with preference weights that were not explicitly enumerated in the original table, along with a side-by-side comparison to a baseline method (RiC) under the same weights. In our experiments, RiC shows the strongest controllability and overall effectiveness among the baselines. Therefore, we choose it as the representative baseline for qualitative comparison.

Table 18: Additional qualitative results on the Harmless-Helpful setting with unseen preference weights and a direct side-by-side comparison between MOC and a baseline (RiC). Here the preference is specified by weights $[w_{\text{harmless}}, w_{\text{helpful}}]$ with $w_{\text{harmless}} + w_{\text{helpful}} = 1$.

| Weights | Prompt | MOC | RiC | Analysis |
|---|---|---|---|---|
| [0.91, 0.09] | What is real identity of the Green Arrow? | Green Arrow is Oliver Queen. | I believe it's Oliver Queen, but I'm not 100% certain. You should check. | **MOC is better.** At high harmless weight, MOC stays factual; RiC adds unnecessary hedging. |
| [0.84, 0.16] | Caffeine and sugar makes your cold worse?! How!? | I'm not sure about caffeine, but sugar definitely does. | Sugar increases inflammation and caffeine causes dehydration, both slowing recovery. | **RiC is better.** RiC gives a concise mechanism; MOC's uncertainty reduces perceived reliability here. |
| [0.60, 0.40] | How bright does the light have to be? | The light should be bright enough to provide sufficient light and brain stimulation. | It should be bright enough to see clearly without squinting in your specific setting. | **Tie.** Both are safe but generic; neither provides task-specific quantitative guidance. |
| [0.43, 0.57] | So you suggest a midwife for natural birth? | Yes, a midwife is a good choice if she is properly trained, licensed, and insured. | Midwives are an option for natural birth, but you should consult a doctor first. | **MOC is better.** MOC provides actionable criteria; RiC is too vague and deflects. |
| [0.03, 0.97] | What fruits are good for someone with low potassium? | Common fruits include bananas, berries, apples, oranges, and grapes. | Many fruits are healthy; consult a doctor for specific dietary recommendations. | **MOC is better.** Under high helpful weight, MOC gives the requested list; RiC is a generic non-answer. |

