# OpenReview forum: "One Model for All: Multi-Objective Controllable Language Models"
_TMLR — Accepted by TMLR_

### Review · Reviewer_1tta · 2025-09-17

**Summary Of Contributions:**

The authors present a rewards-based mechanism for personalising LLM outputs, based upon some (relatively coarse) criteria.

**Additional Comments:**

n/a

**Audience:**

Yes

**Audience Explanation:**

Obviously research in LLMs is incredibly fast moving right now, and there is certainly demand for personalised systems. So on the one hand, this research would easily find an audience. But to me, the broader question is would this audience be able to take anything away from this research? A single model, a single quantization framework, and a very limited exploration regime to me would limit my ability to confidently extract information from this paper, and thus if I was to position myself as an average reader I think that these factors would significantly dilute my interest.

**Broader Impact Concerns:**

See audience interest, but to summarise - an interesting topic, superficially presented well, but with limited experimental evaluation. I appreciate that experimental evaluation on LLMs is difficult to their intrinsic scale, but there's both a limited experimental set, and, in my eyes, a limited evaluation process as well. I actually found Figure 1 to be the most convincing piece of experimental evidence, but yet this does not relate to the fundamental system being studied.

**Claims And Evidence:**

Yes

**Claims Explanation:**

On the one hand, the experimental perspective is strong. Under the experiments conducted, there are clear metric improvements. However, three questions remain: the first is how impactful changes in these measures are? This then follows with how generalisable and extensible would they be, given the single experimental model, and single quantization approach. Finally, when it comes to Figures like Figures 2 and 4, I struggle to see how one could confidently interpret the relative value of one technique over the other via any other approach than the method of looking at it, squinting, and maybe seeing a pattern.

Figures 2 and 4 are particularly interesting, because one could also ask why there is such a strong decoupling between MOC and the other approaches in 2a, that is not replicated in b. The reader is left with no context as to what this decoupling would mean. Nor would one be confident in the limitations of your approach - for example, does it suggest that your technique is not able to approach the low-humor high-helpful regime? If so, does that perhaps speak to a different metric that is not explored by the authors?

The flip side to all of this, however, is that the results are based upon a single model, using a single quantization regime. I'm not highly active in the LLM space, but I do know that in other communities it would be very difficult to convince the readership of impact with such a limited evaluation set. Obviously LLM experimentation is intrinsically more complex, but Figure 1 has demonstrated that there are benefits to toy models, and even a more highly quantized variant of the LLM would make for some level of diversity.

Finally, I'm also curious about how different this approach is to the current approach used by major LLM providers - namely some level of system prompt, that varies between users. Why would one tune for humor vs helpful in the way that the authors have presented, relative to just telling the LLM to be more helpful and less humerus at the system prompt?

**Requested Changes:**

Ideally I would like to see a broader experimental set, and clearer explanations regarding the qualitative and quantitative measures of performance. On a more minor level, while the writing was, on the whole, quite decent, there were a few minor points that I have picked up on that I would request be changed:
- Framing presents efficiency and precision as one contexts objective, and the other to be humor and empathy. Why would one system be optimized for a personal objective on one side, and a cost on the other?
- " limited in providing fully personalized interactions" - statement without citation. Same for multi-objective controllability.
- "due to the high computational costs [associated with LLM training]."
- "is critical [for] enhancing"
- "requires only one training" <-- "requires training only once"
- "and we made it feasible to fine-tune a 7-billion model on a single A6000 GPU with LoRA (Hu et al., 2022)" - this is either poor framing, a violation of double blind, or both?
- "Advantages of MOC." - I like the little breakout for this, however "of MOC" should be bolded, and going straight into a list makes the content more difficult to parse.
- S2.5 - the formatting arround listed items i and ii significantly increases the difficulty to parse. The authors have mixed ideas of lists and sentences in a way that captures the advantages of either.
- Figure 1 caption performs explanations that should, broadly, be in text. And "examined preference weights of "episode reward 1 (wood" are listed below "Preference" is nigh on impenetrable as a form of presentation.
- Figure 2: Numbers in the images are too small. Presentation is incredibly poor - how is one supposed to know how to interpret the value of this?
- The framing around the objectives in S3.1 is not clear. Are humor and helpful alternative hypothesis? What then is harmless and helpful? Is this the alternative to humor and helpful?
- "The quality of solutions, measured using hyper-volumes (as illustrated in Figure 3)" - how does this demonstrate the quality of a solution?
- Theorem 1 is certainly oddly structured as a proof. There's the theorem statement. A general statement. And then the proof. And, at least to me, the line that introduces \pi_{old} is not clear to me, nor is why the next inequality is important? Why are 10/11 and 14/15 defined twice?
- Every time an itemised list is presented as just a block of text - this has to change. It is incredibly difficult to parse out the content, and the format just does not work in the way that you have presented this.

---

### Review · Reviewer_r38W · 2025-12-03

**Summary Of Contributions:**

The authors propose Multi-Objective Control (MOC), a method that enables a model to produce outputs aligned with a user’s preference vector when those preferences are provided online in the prompt. While prior approaches simply optimized a combined objective function created by summing separate preference objectives, MOC achieves controllability by updating both the LLM and the reward head to match the given preference vector.

Strengths

* LLM users can adjust the model’s behavior at inference time without additional training, simply by specifying a preference vector.
* The paper clearly presents comparisons with other models and methods, along with experimental results.

Weaknesses

* The evaluation considers only cases where two out of the three objectives (humor, helpfulness, harmlessness) are active. Therefore, it remains unclear how effective MOC is when multiple objectives coexist simultaneously.
* Modifying the prompt, by prepending the objectives and their preference values, may influence the base LLM’s original capabilities.

**Audience:**

Yes

**Audience Explanation:**

The fact that multiple objectives may coexist is a realistic scenario from the perspective of LLM usage, and this research addresses that issue through MOC. The work also demonstrates, through training, that such objectives can be controlled at inference time, which I believe is a sufficiently interesting result.

**Broader Impact Concerns:**

There are no particular concerns regarding ethical issues.

**Claims And Evidence:**

No

**Claims Explanation:**

The MOC approach in this paper demonstrates strong performance, at least for two objectives, in terms of controllability, solution quality, and generalization to unseen preference vectors compared with other baseline models. However, a major concern is that the model is trained to handle only two objectives, leaving unclear how well the method would scale when extended to multiple objectives. Could authors provide results for a setting where the model is trained on all three given objectives simultaneously, at minimum?

Also regarding the second weakness, what if, instead of inserting objective names and numeric values directly, we leverage the LLM’s inherent instruction-following abilities by framing them in natural language, like this:

```
Please respond to the user while considering the following objectives:
About 0.7 humor,
About 0.3 helpfulness.
```

Wouldn’t this approach allow the model to make better use of the base LLM’s existing capabilities while still benefiting from alignment through preference conditioning?

**Requested Changes:**

It would be great if the paper could be revised primarily with the mentioned weaknesses above.

* Could the notation in the method section be clarified? In several places, new notation appears in the equations but is only defined a few paragraphs later, which makes the explanation somewhat confusing.
* Is it not possible to train on all three presented objectives simultaneously? Is the limitation due to a lack of data? Results with three or more objectives would make the work even more compelling.
* Would adjusting the LLM prompt format lead to better performance? Or does the method already work well without such adjustments?

---

### Review · Reviewer_oAP4 · 2025-12-14

**Summary Of Contributions:**

The authors aim to solve multi-objective optimization problem based on a single LLM. For this, authors introduce Multi-Objective Control (MOC), a unified framework for learning multiple pre-defined objectives based on preference vector. The method integrates multi-objective optimization (MOO) principles into reinforcement learning from human feedback (RLHF) to achieve controllable trade-offs between preferences pre-defined in vector space. The proposed method is computationally efficient, enabling preference optimization of 7B model to be feasible on a single A6000 with LoRA, at the same time, demonstrates superior controllability, solution quality, and robust interpolation on learned preference space, outperforming baselines.

- **Strength**
    - The authors propose a novel and effective multi-objective optimization method that yields a single, controllable LLM. The inclusion of an efficient computational surrogate (Theorem 1) offers a theoretically grounded yet resource-efficient way to approximate multi-objective alignment.
    - MOC efficiently integrates a dynamic preference vector into policy optimization, supporting robust interpolation among diverse behavioral trade-offs without retraining multiple models.
    - The experimental results provides richer evaluation using hyper-volume, Kendall’s tau, and diversity measures, showing consistent controllability and generalization patterns.
- **Weakness**
    - The method directly inputs the preference vector, which belongs to pre-defined vector space, into prompts. While effective from an optimization standpoint, this explicit conditioning fundamentally restrict test-time adaptation to user preference outside the pre-defined preference vector space.
    - The experiments validate interpolation over trained preference vectors but provide limited quantitative evidence for true extrapolation beyond the sampled regions of the preference space. It is unclear how reliably the preference-conditioned control generalizes to real, nuanced human judgments outside the benchmark reward models.
    - The use of numeric preference vectors is theoretically neat but lacks interpretability for end-users. There is minimal discussion on how users might realistically specify preference weights. A clearer bridge between numeric and human-interpretable preference specification (e.g., prompt-based translation) would increase the framework’s usability.

**Audience:**

Yes

**Audience Explanation:**

Since the proposed methods achieves superior Pareto optimality and robust interpolation in the benchmark with efficient training scheme fitted into single A6000, this work could be valuable reference for researchers in this direction.

**Broader Impact Concerns:**

I have no ethical concerns related to this paper.

**Claims And Evidence:**

Yes

**Claims Explanation:**

- **Claim 1.** The claim that MOC offers computational efficiency comparable to single-objective RLHF is credible. The paper provides theoretical support through the surrogate bound (Eq. 12) and practical demonstration via successful fine-tuning of a 7B parameter LLaMA model on a single A6000 GPU.
- **Claim 2.** The authors claim that MOC achieves controllable output behavior that aligns with input preference vectors. This is well-supported by empirical evidence—strong Kendall’s tau correlations, hyper-volume, pair-wise ditance (Table 2-4) and smooth Pareto front transitions in Figures 2 and 4 substantiate the controllability claim.

**Requested Changes:**

- Though the authors says the proposed method is robust under unseen preference, the term “unseen preference” include unnecessary ambiguity. Unseen preference might includes both interpolation and extrapolation in the preference vector space, and this paper provide evidence only for the interpolation context. It would be better clearify the terminology.
- Extend discussion on human interpretability of numeric preference vectors—how human interpretable preference instruction could be compatible with the preference vector?

---

### Decision · Action_Editor_D2dS · 2026-02-22

**Recommendation:** Accept with minor revision

**Additional Comments:**

While the review responses have led to improvements in many aspects and addressed several of the raised concerns, the following issues still require further improvement.

1) The description of the proposed formulation in Section 2.3 is unclear. In particular, starting from Eq. (5), the M preference vectors introduced in Eq. (2) no longer appear in the formulation. It is also unclear what principle motivates the reformulation in Eq. (7), and whether this implies a deviation from the original formulation in Eq. (2). Moreover, there are numerous typos and incorrect equation references throughout this section. Overall, this part requires substantial revision.

2) The baseline descriptions lack sufficient clarity. For example, the fixed preference weights used in MORLHF are not specified, and it is unclear whether MORLHF also conditions the policy on preferences. In addition, the authors should clarify how the preference conditioning in RiC differs from that in MOC (Eq. (4)).

3) An important ablation study related to the proposed objective (Eq. (7)) is missing. Specifically, it would be informative to evaluate the performance when fine-tuning the LLM using only the first objective or only the second objective in the bi-objective formulation of MOC (Eq. (7)), while keeping the same prompt-based preference conditioning for the policy. This ablation would help clarify the individual contributions of each objective.

4) The experimental setup for the target preference alignment results in Table 2 needs to be described more clearly. In particular, it is unclear how the relative ordering of outputs is obtained. If given reward models are used to derive this ordering, it would be more convincing to compare with rankings obtained from human evaluations. In other words, human evaluation results would be more beneficial to support the controllability claims.

5) In Appendices K and L, additional experiments on models beyond LLaMA (e.g., Qwen) would be valuable to demonstrate the generality of the proposed method. Moreover, for the LLaMA-3-8B experiments, it should be clarified whether all baselines are built upon the same LLaMA-3-8B backbone to ensure a fair comparison.

6) The case study in Appendix N is insufficient. More qualitative results are needed, including evaluations on a broader set of inputs, untrained (unseen) preferences, and direct comparisons with baseline results.

Minors

- Fig. 1: more detailed description of Linear PPO would be necessary.
- Pseudocode: P^i = P_j ?
- In section 4.3, what is CPO?

**Audience:**

Yes

**Audience Explanation:**

Controlling LLM outputs according to desired preferences is a practically important problem in real-world LLM usage and is of broad interest to the community. This paper shows that the problem can be addressed effectively and efficiently by fine-tuning existing LLMs in a way that enables easy test-time control, and thus has the potential to draw substantial attention from relevant researchers.

**Claims And Evidence:**

Yes

**Claims Explanation:**

This paper proposes a method, termed Multi-Objective Control (MOC), for fine-tuning a single LLM that can be efficiently steered at inference time via an input preference vector to generate outputs aligned with the specified preferences. In specific, MOC incorporates multiple preference control constraints into an original PPO objective then solves the constrained optimization problem in a computationally efficient manner using a reformulation followed by a surrogate upper bound objective. This leads to improved controllability over individual preferences while simultaneously enabling original policy improvement aligned with the global reward (preference). Extensive experimental results clearly show that the proposed MOC demonstrates superior controllability, as well as higher solution diversity and quality for multiple objectives, compared with baselines including MORLHF and RiC. In addition, MOC shows no performance degradation on unseen preference vectors.

---

> ### Author Response · Authors · 2026-03-02
> **Improvements (1/2)**
>
> Dear Action Editor D2dS,
>
> Thank you again for your detailed and insightful feedback. We have revised the manuscript accordingly. Revisions are highlighted in $\color{#008C5A}{\text{green font}}$. Below we summarize the revisions point-by-point. We would also appreciate your further guidance on the next step in the `TMLR acceptance with minor revision process`.
>
> ---
>
> > **Comment1.1**: The description of the proposed formulation in Section 2.3 is unclear. In particular, starting from Eq. (5), the M preference vectors introduced in Eq. (2) no longer appear in the formulation.
>
> **Revision 1.1:**
>
> - We revised Section 2.3 to clarify that the constraint is imposed over the *finite preference set* $\\{\mathbf{p}^i\\}_{i=1}^M$.
>
> - From Eq. (5) onward, $\mathbf{p}$ denotes a generic element of this set (we omit the superscript $i$ only for readability). We added a short note stating that $\mathbf{p}$ always comes from $\\{\mathbf{p}^i\\}_{i=1}^M$, and we explicitly wrote “for all $i=1,\dots,M$” in the relevant equations.
>
> > **C1.2**: It is also unclear what principle motivates the reformulation in Eq. (7), and whether this implies a deviation from the original formulation in Eq. (2).
>
> **R1.2:**
>
> - We revised Section 2.3 to explain that Eq. (7) is obtained by turning the inequality constraint in Eq. (2) into a hinge-style penalty term (i.e., using the constraint residual as a second objective), which yields an unconstrained multi-objective optimization problem.
> - We explicitly state that Eq. (7) is a **relaxation** (not strictly equivalent to Eq. (2)): any feasible solution to Eq. (2) incurs zero penalty in Eq. (7).
>
> > **C1.3**: Moreover, there are numerous typos and incorrect equation references throughout this section. Overall, this part requires substantial revision.
>
>
> **R1.3:** We proofread Section 2.3 and corrected the typos and equation-reference issues. All edits are highlighted in green in the revised manuscript.
>
>
> > **C2.1**: The baseline descriptions lack sufficient clarity. For example, the fixed preference weights used in MORLHF are not specified, and it is unclear whether MORLHF also conditions the policy on preferences.
>
> **R2.1:** We revised Section 3.1 to specify the fixed preference weights used by MORLHF, i.e., $\{[0.0, 1.0], [0.3, 0.7], [0.5, 0.5], [0.7, 0.3], [1.0, 0.0]\}$. We also clarified that MORLHF does **not** condition the policy on the preference vector. These weights are only used to scalarize multi-dimensional reward-model scores into a single scalar reward for PPO.
>
> > **C2.2**: In addition, the authors should clarify how the preference conditioning in RiC differs from that in MOC (Eq. (4)).
>
> **R2.2:** We clarified that
> - MOC conditions the policy by appending the **preference** vector to the prompt (Eq. (4)), while
> - RiC conditions generation by injecting **rewards** into the prompt, after learning a mapping from preference vectors to rewards (trained with SFT).
>
> This clarification is now stated explicitly in Section 3.1, and we refer readers to **Appendix F** for further discussion.
>
>
> > **C3**: An important ablation study related to the proposed objective (Eq. (7)) is missing. Specifically, it would be informative to evaluate the performance when fine-tuning the LLM using only the first objective or only the second objective in the bi-objective formulation of MOC (Eq. (7)), while keeping the same prompt-based preference conditioning for the policy. This ablation would help clarify the individual contributions of each objective.
>
>
> **R3:** We added Appendix P (Ablation Study) in the revised manuscript.
>
> | Fine-tuning objective | Hyper-volume ($\uparrow$) | Kendall's tau ($\uparrow$) | p-value ($\downarrow$) |
> |---|---:|---:|---:|
> | MOC (ours; full bi-objective) | 10.905 | 0.9636 | 0.000001 |
> | Only MO objective (w/o controllability) | 6.334 | 0.5636 | 0.016541 |
> | Only controllability objective (w/o MO optimization) | 1.869 | 0.0182 | 1.000000 |
>
> Optimizing only one objective either degrades controllability or fails to improve solution quality, while the full bi-objective formulation achieves both, further validating the effectiveness of our MOC objective design.

---

> ### Author Response · Authors · 2026-03-02
> **Improvements (2/2)**
>
> > **C4.1**: The experimental setup for the target preference alignment results in Table 2 needs to be described more clearly. In particular, it is unclear how the relative ordering of outputs is obtained.
>
> **R4.1:** We revised Section 3.2 to clarify that the relative ordering of outputs is obtained by ranking outputs using a projection/angle-based score computed from their reward vectors, and we now explicitly direct readers to the existing computation details in **Appendix I** (Kendall’s Tau Computation Details).
>
> > **C4.2**: If given reward models are used to derive this ordering, it would be more convincing to compare with rankings obtained from human evaluations. In other words, human evaluation results would be more beneficial to support the controllability claims.
>
> **R4.2:** We agree that human evaluation would provide stronger evidence. We added a brief discussion in **Section 3.1** clarifying that we use reward-model-based rankings as a widely adopted proxy in RLHF, since the reward models are trained from human preference data.
>
>
> > **C5**: In Appendices K and L, additional experiments on models beyond LLaMA (e.g., Qwen) would be valuable to demonstrate the generality of the proposed method. Moreover, for the LLaMA-3-8B experiments, it should be clarified whether all baselines are built upon the same LLaMA-3-8B backbone to ensure a fair comparison.
>
> **R5**:
>
> - We added **Appendix K.1 (Qwen2.5-7B)** to demonstrate generality beyond LLaMA: on HH-RLHF, MOC achieves hyper-volume 10.905 and Kendall’s $\tau=0.9636$ ($p=10^{-6}$), outperforming the baselines reported there (e.g., RiC: 9.257; MORLHF: 9.043). Appendix K.1 further shows that MOC remains effective and achieves strong performance across other backbones.
> - We also added a brief clarification in Appendix K about the backbone settings for fair comparison.
>
>
> > **C6**: The case study in Appendix N is insufficient. More qualitative results are needed, including evaluations on a broader set of inputs, untrained (unseen) preferences, and direct comparisons with baseline results
>
>
> **R6:** We added additional qualitative results in Appendix N (Case Study) on a broader set of prompts and untrained (unseen) preference weights, including a direct side-by-side comparison between MOC and RiC.
>
>
> > **C7.1** Fig. 1: more detailed description of Linear PPO would be necessary.
>
> **R7.1:** We added a more detailed description of Linear PPO in the caption of Fig. 1, clarifying that it optimizes PPO on a linearly scalarized reward (using weights $[0.1,0.9],[0.2,0.8],\cdots,[0.8,0.2],[0.9,0.1]$).
>
>
> > **C7.2** Pseudocode: P^i = P_j?
>
> **R7.2:** We revised the pseudocode (Appendix D: Pseudocode) to avoid any ambiguity between the set index and the mini-batch index: we treat $\mathbb{P}$ as a finite preference-vector set and explicitly state that the mini-batch $\\{\mathbf{p}\_j\\}_{j=1}^B$ is sampled from $\mathbb{P}$ (where subscript $j$ only indexes the mini-batch).
>
>
> > **C7.3** In section 4.3, what is CPO?
>
> **R7.3:** CPO refers to the paper: Controllable Preference Optimization: Toward Controllable Multi-Objective Alignment (EMNLP 2024). We clarified this by adding a citation where CPO is mentioned.
>
> ---
>
> Thank you again for your time and guidance.
>
> Sincerely,
> The authors